# Collaboratively Learning Preferences from Ordinal Data

**Sewoong Oh** , **Kiran K. Thekumparampil**
University of Illinois at Urbana-Champaign
{swoh,thekump2}@illinois.edu

**Jiaming Xu**
The Wharton School, UPenn
jiamingx@wharton.upenn.edu

## Abstract

In personalized recommendation systems, it is important to predict preferences of a user on items that have not been seen by that user yet. Similarly, in revenue management, it is important to predict outcomes of comparisons among those items that have never been compared so far. The MultiNomial Logit model, a popular discrete choice model, captures the structure of the hidden preferences with a low-rank matrix. In order to predict the preferences, we want to learn the underlying model from noisy observations of the low-rank matrix, collected as revealed preferences in various forms of ordinal data. A natural approach to learn such a model is to solve a convex relaxation of nuclear norm minimization. We present the convex relaxation approach in two contexts of interest: collaborative ranking and bundled choice modeling. In both cases, we show that the convex relaxation is minimax optimal. We prove an upper bound on the resulting error with finite samples, and provide a matching information-theoretic lower bound.

## 1 Introduction

In recommendation systems and revenue management, it is important to predict preferences on items that have not been seen by a user or predict outcomes of comparisons among those that have never been compared. Predicting such hidden preferences would be hopeless without further assumptions on the structure of the preference. Motivated by the success of matrix factorization models on collaborative filtering applications, we model hidden preferences with low-rank matrices to collaboratively learn preference matrices from ordinal data. This paper considers the following scenarios:

- *Collaborative ranking.* Consider an online market that collects each user's preference as a ranking over a subset of items that are 'seen' by the user. Such data can be obtained by directly asking to compare some items, or by indirectly tracking online activities on which items are viewed, how much time is spent on the page, or how the user rated the items. In order to make personalized recommendations, we want a model which (a) captures how users who prefer similar items are also likely to have similar preferences on unseen items, (b) predicts which items a user might prefer, by learning from such ordinal data.

- *Bundled choice modeling.* Discrete choice models describe how a user makes decisions on what to purchase. Typical choice models assume the willingness to buy an item is independent of what else the user bought. In many cases, however, we make 'bundled' purchases: we buy particular ingredients together for one recipe or we buy two connecting flights. One choice (the first flight) has a significant impact on the other (the connecting flight). In order to optimize the assortment (which flight schedules to offer) for maximum expected revenue, it is crucial to accurately predict the willingness of the consumers to purchase, based on past history. We consider a case where there are two types of products (e.g. jeans and shirts), and want $(a)$ a model that captures such interacting preferences for pairs of items, one from each category; and $(b)$ to predict the consumer's choice probabilities on pairs of items, by learning such models from past purchase history.

We use a discrete choice model known as MultiNomial Logit (MNL) model [1] (described in Section 2.1) to represent the preferences. In collaborative ranking context, MNL uses a low-rank matrix to represent the hidden preferences of the users. Each row corresponds to a user's preference over all the items, and when presented with a subset of items the user provides a ranking over those items, which is a noisy version of the hidden true preference. The low-rank assumption naturally captures the similarities among users and items, by representing each on a low-dimensional space. In bundled choice modeling context, the low-rank matrix now represents how pairs of items are matched. Each row corresponds to an item from the first category and each column corresponds to an item from the second category. An entry in the matrix represents how much the pair is preferred by a randomly chosen user from a pool of users. Notice that in this case we do not model individual preferences, but the preference of the whole population. The purchase history of the population is the record of which pair was chosen among a subsets of items that were presented, which is again a noisy version of the hidden true preference. The low-rank assumption captures the similarities and dis-similarities among the items in the same category and the interactions across categories.

**Contribution.** A natural approach to learn such a low-rank model, from noisy observations, is to solve a convex relaxation of nuclear norm minimization (described in Section 2.2), since nuclear norm is the tightest convex surrogate for the rank function. We present such an approach for learning the MNL model from ordinal data, in two contexts: collaborative ranking and bundled choice modeling. In both cases, we analyze the sample complexity of the algorithm, and provide an upper bound on the resulting error with finite samples. We prove minimax-optimality of our approach by providing a matching information-theoretic lower bound (up to a poly-logarithmic factor). Technically, we utilize the Random Utility Model (RUM) [2, 3, 4] interpretation (outlined in Section 2.1) of the MNL model to prove both the upper bound and the fundamental limit, which could be of interest to analyzing more general class of RUMs.

**Related work.** In the context of collaborative ranking, MNL models have been proposed to model partial rankings from a pool of users. Recently, there has been new algorithms and analyses of those algorithms to learn MNL models from samples, in the case when each user provides pair-wise comparisons [5, 6]. [6] proposes solving a convex relaxation of maximizing the likelihood over matrices with bounded nuclear norm. It is shown that this approach achieves statistically optimal generalization error rate, instead of Frobenius norm error that we analyze. Our analysis techniques are inspired by [5], which proposed the convex relaxation for learning MNL, but when the users provide only pair-wise comparisons. In this paper, we generalize the results of [5] by analyzing more general sampling models beyond pairwise comparisons.

The remainder of the paper is organized as follows. In Section 2, we present the MNL model and propose a convex relaxation for learning the model, in the context of collaborative ranking. We provide theoretical guarantees for collaborative ranking in Section 3. In Section 4, we present the problem statement for bundled choice modeling, and analyze a similar convex relaxation approach.

**Notations.** We use $\|\|A\|\|_{\mathrm{F}}$ and $\|\|A\|\|_{\infty}$ to denote the Frobenius norm and the $\ell_\infty$ norm, $\|\|A\|\|_{\mathrm{nuc}} = \sum_i \sigma_i(A)$ to denote the nuclear norm where $\sigma_i(A)$ denote the $i$-th singular value, and $\|\|A\|\|_2 = \sigma_1(A)$ for the spectral norm. We use $\langle\langle u, v \rangle\rangle = \sum_i u_i v_i$ and $\|u\|$ to denote the inner product and the Euclidean norm. All ones vector is denoted by $\mathbb{1}$ and $\mathbb{I}(A)$ is the indicator function of the event $A$. The set of the fist $N$ integers are denoted by $[N] = \{1, \ldots, N\}$.

## 2 Model and Algorithm

In this section, we present a discrete choice modeling for collaborative ranking, and propose an inference algorithm for learning the model from ordinal data.

### 2.1 MultiNomial Logit (MNL) model for comparative judgment

In collaborative ranking, we want to model how people who have similar preferences on a subset of items are likely to have similar tastes on other items as well. When users provide ratings, as in collaborative filtering applications, matrix factorization models are widely used since the low-rank structure captures the similarities between users. When users provide ordered preferences, we use a discrete choice model known as MultiNomial Logit (MNL) [1] model that has a similar low-rank structure that captures the similarities between users and items.

Let $\Theta^*$ be the $d_1 \times d_2$ dimensional matrix capturing the preference of $d_1$ users on $d_2$ items, where the rows and columns correspond to users and items, respectively. Typically, $\Theta^*$ is assumed to be low-rank, having a rank $r$ that is much smaller than the dimensions. However, in the following we allow a more general setting where $\Theta^*$ might be only approximately low rank. When a user $i$ is presented with a set of alternatives $S_i \subseteq [d_2]$, she reveals her preferences as a ranked list over those items. To simplify the notations we assume all users compare the same number $k$ of items, but the analysis naturally generalizes to the case when the size might differ from a user to a user. Let $v_{i,\ell} \in S_i$ denote the (random) $\ell$-th best choice of user $i$. Each user gives a ranking, independent of other users' rankings, from

$$\mathbb{P}\{v_{i,1}, \ldots, v_{i,k}\} = \prod_{\ell=1}^{k} \frac{e^{\Theta^*_{i,v_{i,\ell}}}}{\sum_{j \in S_{i,\ell}} e^{\Theta^*_{i,j}}} , \tag{1}$$

where with $S_{i,\ell} \equiv S_i \setminus \{v_{i,1}, \ldots, v_{i,\ell-1}\}$ and $S_{i,1} \equiv S_i$. For a user $i$, the $i$-th row of $\Theta^*$ represents the underlying preference vector of the user, and the more preferred items are more likely to be ranked higher. The probabilistic nature of the model captures the noise in the revealed preferences.

The random utility model (RUM), pioneered by [2, 3, 4], describes the choices of users as manifestations of the underlying utilities. The MNL models is a special case of RUM where each decision maker and each alternative are represented by a $r$-dimensional feature vectors $u_i$ and $v_j$ respectively, such that $\Theta^*_{ij} = \langle\!\langle u_i, v_j \rangle\!\rangle$, resulting in a low-rank matrix. When presented with a set of alternatives $S_i$, the decision maker $i$ ranks the alternatives according to their random utility drawn from

$$U_{ij} = \langle\!\langle u_i, v_j \rangle\!\rangle + \xi_{ij} , \tag{2}$$

for item $j$, where $\xi_{ij}$ follow the standard Gumbel distribution. Intuitively, this provides a justification for the MNL model as modeling the decision makers as rational being, seeking to maximize utility. Technically, this RUM interpretation plays a crucial role in our analysis, in proving restricted strong convexity in Appendix A.5 and also in proving fundamental limit in Appendix C.

There are a few cases where the Maximum Likelihood (ML) estimation for RUM is tractable. One notable example is the Plackett-Luce (PL) model, which is a special case of the MNL model where $\Theta^*$ is rank-one and all users have the same features. PL model has been widely applied in econometrics [1], analyzing elections [7], and machine learning [8]. Efficient inference algorithms has been proposed [9, 10, 11], and the sample complexity has been analyzed for the MLE [12] and for the Rank Centrality [13]. Although PL is quite restrictive, in the sense that it assumes all users share the same features, little is known about inference in RUMs beyond PL. Recently, to overcome such a restriction, mixed PL models have been studied, where $\Theta^*$ is rank-$r$ but there are only $r$ classes of users and all users in the same class have the same features. Efficient inference algorithms with provable guarantees have been proposed by applying recent advances in tensor decomposition methods [14, 15], directly clustering the users [16, 17], or using sampling methods [18]. However, this mixture PL is still restrictive, and both clustering and tensor based approaches rely heavily on the fact that the distribution is a "mixture" and require additional incoherence assumptions on $\Theta^*$. For more general models, efficient inference algorithms have been proposed [19] but no performance guarantee is known for finite samples. Although the MLE for the general MNL model in (1) is intractable, we provide a polynomial-time inference algorithm with provable guarantees.

## 2.2 Nuclear norm minimization

Assuming $\Theta^*$ is well approximated by a low-rank matrix, we estimate $\Theta^*$ by solving the following convex relaxation given the observed preference in the form of ranked lists $\{(v_{i,1}, \ldots, v_{i,k})\}_{i \in [d_1]}$.

$$\widehat{\Theta} \in \arg\min_{\Theta \in \Omega} \mathcal{L}(\Theta) + \lambda \|\|\Theta\|\|_{\mathrm{nuc}}, \tag{3}$$

where the (negative) log likelihood function according to (1) is

$$\mathcal{L}(\Theta) = -\frac{1}{k\,d_1} \sum_{i=1}^{d_1} \sum_{\ell=1}^{k} \left( \langle\!\langle \Theta, e_i e_{v_{i,\ell}}^T \rangle\!\rangle - \log \left( \sum_{j \in S_{i,\ell}} \exp\left( \langle\!\langle \Theta, e_i e_j^T \rangle\!\rangle \right) \right) \right) , \tag{4}$$

with $S_i = \{v_{i,1}, \ldots, v_{i,k}\}$ and $S_{i,\ell} \equiv S_i \setminus \{v_{i,1}, \ldots, v_{i,\ell-1}\}$, and appropriately chosen set $\Omega$ defined in (7). Since nuclear norm is a tight convex surrogate for the rank, the above optimization searches

for a low-rank solution that maximizes the likelihood. Nuclear norm minimization has been widely used in rank minimization problems [20], but provable guarantees typically exists only for quadratic loss function $\mathcal{L}(\Theta)$ [21, 22]. Our analysis extends such analysis techniques to identify the conditions under which restricted strong convexity is satisfied for a convex loss function that is not quadratic.

## 3  Collaborative ranking from $k$-wise comparisons

We first provide background on the MNL model, and then present main results on the performance guarantees. Notice that the distribution (1) is independent of shifting each row of $\Theta^*$ by a constant. Hence, there is an equivalent class of $\Theta^*$ that gives the same distributions for the ranked lists:

$$[\Theta^*] = \{A \in \mathbb{R}^{d_1 \times d_2} \mid A = \Theta^* + u \mathbb{1}^T \text{ for some } u \in \mathbb{R}^{d_1}\} . \tag{5}$$

Since we can only estimate $\Theta^*$ up to this equivalent class, we search for the one whose rows sum to zero, i.e. $\sum_{j \in [d_2]} \Theta^*_{i,j} = 0$ for all $i \in [d_1]$. Let $\alpha \equiv \max_{i,j_1,j_2} |\Theta^*_{ij_1} - \Theta^*_{ij_2}|$ denote the dynamic range of the underlying $\Theta^*$, such that when $k$ items are compared, we always have

$$\frac{1}{k} e^{-\alpha} \;\leq\; \frac{1}{1 + (k-1)e^{\alpha}} \;\leq\; \mathbb{P}\{v_{i,1} = j\} \;\leq\; \frac{1}{1 + (k-1)e^{-\alpha}} \;\leq\; \frac{1}{k} e^{\alpha} , \tag{6}$$

for all $j \in S_i$, all $S_i \subseteq [d_2]$ satisfying $|S_i| = k$ and all $i \in [d_1]$. We do not make any assumptions on $\alpha$ other than that $\alpha = O(1)$ with respect to $d_1$ and $d_2$. The purpose of defining the dynamic range in this way is that we seek to characterize how the error scales with $\alpha$. Given this definition, we solve the optimization in (3) over

$$\Omega_\alpha = \left\{ A \in \mathbb{R}^{d_1 \times d_2} \,\middle|\, \|A\|_\infty \leq \alpha, \text{ and } \forall i \in [d_1] \text{ we have } \sum_{j \in [d_2]} A_{ij} = 0 \right\} . \tag{7}$$

While in practice we do not require the $\ell_\infty$ norm constraint, we need it for the analysis. For a related problem of matrix completion, where the loss $\mathcal{L}(\theta)$ is quadratic, either a similar condition on $\ell_\infty$ norm is required or a different condition on incoherence is required.

### 3.1  Performance guarantee

We provide an upper bound on the resulting error of our convex relaxation, when a *multi-set* of items $S_i$ presented to user $i$ is drawn uniformly at random with replacement. Precisely, for a given $k$, $S_i = \{j_{i,1}, \ldots, j_{i,k}\}$ where $j_{i,\ell}$'s are independently drawn uniformly at random over the $d_2$ items. Further, if an item is sampled more than once, i.e. if there exists $j_{i,\ell_1} = j_{i,\ell_2}$ for some $i$ and $\ell_1 \neq \ell_2$, then we assume that the user treats these two items as if they are two distinct items with the same MNL weights $\Theta^*_{i,j_{i,\ell_1}} = \Theta^*_{i,j_{i,\ell_2}}$. The resulting preference is therefore always over $k$ items (with possibly multiple copies of the same item), and distributed according to (1). For example, if $k = 3$, it is possible to have $S_i = \{j_{i,1} = 1, j_{i,2} = 1, j_{i,3} = 2\}$, in which case the resulting ranking can be $(v_{i,1} = j_{i,1}, v_{i,2} = j_{i,3}, v_{i,3} = j_{i,2})$ with probability $(e^{\Theta_{i,1}})/(2\, e^{\Theta_{i,1}} + e^{\Theta_{i,2}}) \times (e^{\Theta_{i,2}})/(e^{\Theta_{i,1}} + e^{\Theta_{i,2}})$. Such sampling with replacement is necessary for the analysis, where we require independence in the choice of the items in $S_i$ in order to apply the symmetrization technique (e.g. [23]) to bound the expectation of the deviation (cf. Appendix A.5). Similar sampling assumptions have been made in existing analyses on learning low-rank models from noisy observations, e.g. [22]. Let $d \equiv (d_1 + d_2)/2$, and let $\sigma_j(\Theta^*)$ denote the $j$-th singular value of the matrix $\Theta^*$. Define

$$\lambda_0 \;\equiv\; e^{2\alpha} \sqrt{\frac{d_1 \log d + d_2 (\log d)^2 (\log 2d)^4}{k\, d_1^2\, d_2}} \;.$$

**Theorem 1.** *Under the described sampling model, assume $24 \leq k \leq \min\{d_1^2 \log d, (d_1^2 + d_2^2)/(2d_1) \log d, (1/e)\, d_2(4 \log d_2 + 2 \log d_1)\}$, and $\lambda \in [480\lambda_0, c_0\lambda_0]$ with any constant $c_0 = O(1)$ larger than 480. Then, solving the optimization (3) achieves*

$$\frac{1}{d_1 d_2} \left\|\widehat{\Theta} - \Theta^*\right\|_F^2 \;\leq\; 288\sqrt{2}\, e^{4\alpha} c_0 \lambda_0 \sqrt{r} \left\|\widehat{\Theta} - \Theta^*\right\|_F + 288 e^{4\alpha} c_0 \lambda_0 \sum_{j=r+1}^{\min\{d_1, d_2\}} \sigma_j(\Theta^*) , \tag{8}$$

*for any $r \in \{1, \ldots, \min\{d_1, d_2\}\}$ with probability at least $1 - 2d^{-3} - d_2^{-3}$ where $d = (d_1 + d_2)/2$.*

A proof is provided in Appendix A. The above bound shows a natural splitting of the error into two terms, one corresponding to the *estimation error* for the rank-$r$ component and the second one corresponding to the *approximation error* for how well one can approximate $\Theta^*$ with a rank-$r$ matrix. This bound holds for all values of $r$ and one could potentially optimize over $r$. We show such results in the following corollaries.

**Corollary 3.1 (Exact low-rank matrices).** *Suppose $\Theta^*$ has rank at most $r$. Under the hypotheses of Theorem 1, solving the optimization (3) with the choice of the regularization parameter $\lambda \in [480\lambda_0, c_0\lambda_0]$ achieves with probability at least $1 - 2d^{-3} - d_2^{-3}$,*

$$\frac{1}{\sqrt{d_1 d_2}} \left\| \widehat{\Theta} - \Theta^* \right\|_{\mathrm{F}} \le 288\sqrt{2}e^{6\alpha}c_0 \sqrt{\frac{r(d_1 \log d + d_2 (\log d)^2 (\log 2d)^4)}{k \, d_1}} \; . \tag{9}$$

The number of entries is $d_1 d_2$ and we rescale the Frobenius norm error appropriately by $1/\sqrt{d_1 d_2}$. When $\Theta^*$ is a rank-$r$ matrix, then the degrees of freedom in representing $\Theta^*$ is $r(d_1 + d_2) - r^2 = O(r(d_1 + d_2))$. The above theorem shows that the total number of samples, which is $(k \, d_1)$, needs to scale as $O(rd_1(\log d) + rd_2 (\log d)^2 (\log 2d)^4$ in order to achieve an arbitrarily small error. This is only poly-logarithmic factor larger than the degrees of freedom. In Section 3.2, we provide a lower bound on the error directly, that matches the upper bound up to a logarithmic factor.

The dependence on the dynamic range $\alpha$, however, is sub-optimal. It is expected that the error increases with $\alpha$, since the $\Theta^*$ scales as $\alpha$, but the exponential dependence in the bound seems to be a weakness of the analysis, as seen from numerical experiments in the right panel of Figure 1. Although the error increase with $\alpha$, numerical experiments suggests that it only increases at most linearly. However, tightening the scaling with respect to $\alpha$ is a challenging problem, and such sub-optimal dependence is also present in existing literature for learning even simpler models, such as the Bradley-Terry model [13] or the Plackett-Luce model [12], which are special cases of the MNL model studied in this paper. A practical issue in achieving the above rate is the choice of $\lambda$, since the dynamic range $\alpha$ is not known in advance. Figure 1 illustrates that the error is not sensitive to the choice of $\lambda$ for a wide range.

Another issue is that the underlying matrix might not be exactly low rank. It is more realistic to assume that it is approximately low rank. Following [22] we formalize this notion with "$\ell_q$-ball" of matrices defined as

$$\mathbb{B}_q(\rho_q) \equiv \left\{ \Theta \in \mathbb{R}^{d_1 \times d_2} \mid \sum_{j \in [\min\{d_1, d_2\}]} |\sigma_j(\Theta^*)|^q \le \rho_q \right\} . \tag{10}$$

When $q = 0$, this is a set of rank-$\rho_0$ matrices. For $q \in (0, 1]$, this is set of matrices whose singular values decay relatively fast. Optimizing the choice of $r$ in Theorem 1, we get the following result.

**Corollary 3.2 (Approximately low-rank matrices).** *Suppose $\Theta^* \in \mathbb{B}_q(\rho_q)$ for some $q \in (0, 1]$ and $\rho_q > 0$. Under the hypotheses of Theorem 1, solving the optimization (3) with the choice of the regularization parameter $\lambda \in [480\lambda_0, c_0\lambda_0]$ achieves with probability at least $1 - 2d^{-3}$,*

$$\frac{1}{\sqrt{d_1 d_2}} \left\| \widehat{\Theta} - \Theta^* \right\|_{\mathrm{F}} \le \frac{2\sqrt{\rho_q}}{\sqrt{d_1 d_2}} \left( 288\sqrt{2}c_0 e^{6\alpha} \sqrt{\frac{d_1 d_2 (d_1 \log d + d_2 (\log d)^2 (\log 2d)^2)}{k \, d_1}} \right)^{\frac{2-q}{2}} . \tag{11}$$

This is a strict generalization of Corollary 3.1. For $q = 0$ and $\rho_0 = r$, this recovers the exact low-rank estimation bound up to a factor of two. For approximate low-rank matrices in an $\ell_q$-ball, we lose in the error exponent, which reduces from one to $(2 - q)/2$. A proof of this Corollary is provided in Appendix B.

The left panel of Figure 1 confirms the scaling of the error rate as predicted by Corollary 3.1. The lines merge to a single line when the sample size is rescaled appropriately. We make a choice of $\lambda = (1/2)\sqrt{(\log d)/(kd^2)}$, This choice is independent of $\alpha$ and is smaller than proposed in Theorem 1. We generate random rank-$r$ matrices of dimension $d \times d$, where $\Theta^* = UV^T$ with $U \in \mathbb{R}^{d \times r}$ and $V \in \mathbb{R}^{d \times r}$ entries generated i.i.d from uniform distribution over $[0, 1]$. Then the

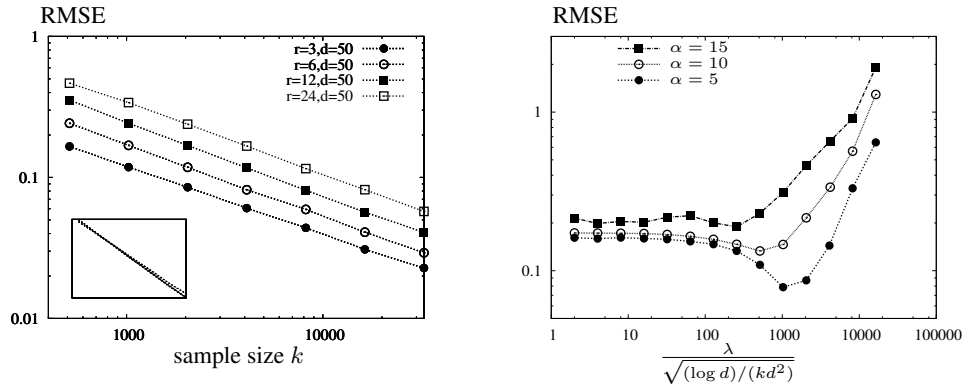

Figure 1: The (rescaled) RMSE scales as $\sqrt{r(\log d)/k}$ as expected from Corollary 3.1 for fixed $d = 50$ (left). In the inset, the same data is plotted versus rescaled sample size $k/(r \log d)$. The (rescaled) RMSE is stable for a broad range of $\lambda$ and $\alpha$ for fixed $d = 50$ and $r = 3$ (right).

row-mean is subtracted form each row, and then the whole matrix is scaled such that the largest entry is $\alpha = 5$. Note that this operation does not increase the rank of the matrix $\Theta$. This is because this de-meaning can be written as $\Theta - \Theta \mathbb{1}\mathbb{1}^T/d_2$ and both terms in the operation are of the same column space as $\Theta$ which is of rank $r$. The root mean squared error (RMSE) is plotted where $\mathrm{RMSE} = (1/d)\|\Theta^* - \widehat{\Theta}\|_{\mathrm{F}}$. We implement and solve the convex optimization (3) using proximal gradient descent method as analyzed in [24]. The right panel in Figure 1 illustrates that the actual error is insensitive to the choice of $\lambda$ for a broad range of $\lambda \in [\sqrt{(\log d)/(kd^2)}, 2^8\sqrt{(\log d)/(kd^2)}]$, after which it increases with $\lambda$.

## 3.2 Information-theoretic lower bound for low-rank matrices

For a polynomial-time algorithm of convex relaxation, we gave in the previous section a bound on the achievable error. We next compare this to the fundamental limit of this problem, by giving a lower bound on the achievable error by any algorithm (efficient or not). A simple parameter counting argument indicates that it requires the number of samples to scale as the degrees of freedom i.e., $kd_1 \propto r(d_1 + d_2)$, to estimate a $d_1 \times d_2$ dimensional matrix of rank $r$. We construct an appropriate packing over the set of low-rank matrices with bounded entries in $\Omega_\alpha$ defined as (7), and show that no algorithm can accurately estimate the true matrix with high probability using the generalized Fano's inequality. This provides a constructive argument to lower bound the minimax error rate, which in turn establishes that the bounds in Theorem 1 is sharp up to a logarithmic factor, and proves no other algorithm can significantly improve over the nuclear norm minimization.

**Theorem 2.** *Suppose $\Theta^*$ has rank $r$. Under the described sampling model, for large enough $d_1$ and $d_2 \geq d_1$, there is a universal numerical constant $c > 0$ such that*

$$\inf_{\widehat{\Theta}} \sup_{\Theta^* \in \Omega_\alpha} \mathbb{E}\left[\frac{1}{\sqrt{d_1 d_2}}\left\|\widehat{\Theta} - \Theta^*\right\|_{\mathrm{F}}\right] \geq c \min\left\{\alpha e^{-\alpha}\sqrt{\frac{r\,d_2}{k\,d_1}}, \frac{\alpha d_2}{\sqrt{d_1 d_2 \log d}}\right\}, \quad (12)$$

*where the infimum is taken over all measurable functions over the observed ranked lists $\{(v_{i,1}, \ldots, v_{i,k})\}_{i \in [d_1]}$.*

A proof of this theorem is provided in Appendix C. The term of primary interest in this bound is the first one, which shows the scaling of the (rescaled) minimax rate as $\sqrt{r(d_1 + d_2)/(kd_1)}$ (when $d_2 \geq d_1$), and matches the upper bound in (8). It is the dominant term in the bound whenever the number of samples is larger than the degrees of freedom by a logarithmic factor, i.e., $kd_1 > r(d_1 + d_2)\log d$, ignoring the dependence on $\alpha$. This is a typical regime of interest, where the sample size is comparable to the latent dimension of the problem. In this regime, Theorem 2 establishes that the upper bound in Theorem 1 is minimax-optimal up to a logarithmic factor in the dimension $d$.

# 4 Choice modeling for bundled purchase history

In this section, we use the MNL model to study another scenario of practical interest: choice modeling from bundled purchase history. In this setting, we assume that we have bundled purchase history data from $n$ users. Precisely, there are two categories of interest with $d_1$ and $d_2$ alternatives in each category respectively. For example, there are $d_1$ tooth pastes to choose from and $d_2$ tooth brushes to choose from. For the $i$-th user, a subset $S_i \subseteq [d_1]$ of alternatives from the first category is presented along with a subset $T_i \subseteq [d_2]$ of alternatives from the second category. We use $k_1$ and $k_2$ to denote the number of alternatives presented to a single user, i.e. $k_1 = |S_i|$ and $k_2 = |T_i|$, and we assume that the number of alternatives presented to each user is fixed, to simplify notations. Given these sets of alternatives, each user makes a 'bundled' purchase and we use $(u_i, v_i)$ to denote the bundled pair of alternatives (e.g. a tooth brush and a tooth paste) purchased by the $i$-th user. Each user makes a choice of the best alternative, independent of other users's choices, according to the MNL model as

$$\mathbb{P}\left\{(u_i, v_i) = (j_1, j_2)\right\} \quad = \quad \frac{e^{\Theta^*_{j_1, j_2}}}{\sum_{j'_1 \in S_i, j'_2 \in T_i} e^{\Theta^*_{j'_1, j'_2}}} \ , \tag{13}$$

for all $j_1 \in S_i$ and $j_2 \in T_i$. The distribution (13) is independent of shifting all the values of $\Theta^*$ by a constant. Hence, there is an equivalent class of $\Theta^*$ that gives the same distribution for the choices: $[\Theta^*] \equiv \{A \in \mathbb{R}^{d_1 \times d_2} \,|\, A = \Theta^* + c\mathbb{1}\mathbb{1}^T \text{ for some } c \in \mathbb{R}\}$ . Since we can only estimate $\Theta^*$ up to this equivalent class, we search for the one that sum to zero, i.e. $\sum_{j_1 \in [d_1], j_2 \in [d_2]} \Theta^*_{j_1, j_2} = 0$. Let $\alpha = \max_{j_1, j'_1 \in [d_1], j_2, j'_2 \in [d_2]} |\Theta^*_{j_1, j_2} - \Theta^*_{j'_1, j'_2}|$, denote the dynamic range of the underlying $\Theta^*$, such that when $k_1 \times k_2$ alternatives are presented, we always have

$$\frac{1}{k_1 k_2} e^{-\alpha} \ \leq \ \mathbb{P}\left\{(u_i, v_i) = (j_1, j_2)\right\} \ \leq \ \frac{1}{k_1 k_2} e^{\alpha} \ , \tag{14}$$

for all $(j_1, j_2) \in S_i \times T_i$ and for all $S_i \subseteq [d_1]$ and $T_i \subseteq [d_2]$ such that $|S_i| = k_1$ and $|T_i| = k_2$. We do not make any assumptions on $\alpha$ other than that $\alpha = O(1)$ with respect to $d_1$ and $d_2$. Assuming $\Theta^*$ is well approximate by a low-rank matrix, we solve the following convex relaxation, given the observed bundled purchase history $\{(u_i, v_i, S_i, T_i)\}_{i \in [n]}$:

$$\widehat{\Theta} \ \in \ \arg\min_{\Theta \in \Omega'_\alpha} \mathcal{L}(\Theta) + \lambda \||\Theta\||_{\text{nuc}} \ , \tag{15}$$

where the (negative) log likelihood function according to (13) is

$$\mathcal{L}(\Theta) \ = \ -\frac{1}{n} \sum_{i=1}^{n} \left( \langle\!\langle \Theta, e_{u_i} e_{v_i}^T \rangle\!\rangle - \log\left( \sum_{j_1 \in S_i, j_2 \in T_i} \exp\left( \langle\!\langle \Theta, e_{j_1} e_{j_2}^T \rangle\!\rangle \right) \right) \right), \text{ and} \tag{16}$$

$$\Omega'_\alpha \ \equiv \ \left\{ A \in \mathbb{R}^{d_1 \times d_2} \,\Big|\, \||A\||_\infty \leq \alpha, \text{ and } \sum_{j_1 \in [d_1], j_2 \in [d_2]} A_{j_1, j_2} = 0 \right\}. \tag{17}$$

Compared to collaborative ranking, $(a)$ rows and columns of $\Theta^*$ correspond to an alternative from the first and second category, respectively; $(b)$ each sample corresponds to the purchase choice of a user which follow the MNL model with $\Theta^*$; $(c)$ each person is presented subsets $S_i$ and $T_i$ of items from each category; $(d)$ each sampled data represents the most preferred bundled pair of alternatives.

## 4.1 Performance guarantee

We provide an upper bound on the error achieved by our convex relaxation, when the *multi-set* of alternatives $S_i$ from the first category and $T_i$ from the second category are drawn uniformly at random with replacement from $[d_1]$ and $[d_2]$ respectively. Precisely, for given $k_1$ and $k_2$, we let $S_i = \{j_{1,1}^{(i)}, \ldots, j_{1,k_1}^{(i)}\}$ and $T_i = \{j_{2,1}^{(i)}, \ldots, j_{2,k_2}^{(i)}\}$, where $j_{1,\ell}^{(i)}$'s and $j_{2,\ell}^{(i)}$'s are independently drawn uniformly at random over the $d_1$ and $d_2$ alternatives, respectively. Similar to the previous section, this sampling with replacement is necessary for the analysis. Define

$$\lambda_1 = \sqrt{\frac{e^{2\alpha} \max\{d_1, d_2\} \log d}{n \, d_1 \, d_2}} \ . \tag{18}$$

**Theorem 3.** *Under the described sampling model, assume* $16e^{2\alpha} \min\{d_1, d_2\} \log d \leq n \leq \min\{d^5, k_1 k_2 \max\{d_1^2, d_2^2\}\} \log d$, *and* $\lambda \in [8\lambda_1, c_1\lambda_1]$ *with any constant* $c_1 = O(1)$ *larger than* $\max\{8, 128/\sqrt{\min\{k_1, k_2\}}\}$. *Then, solving the optimization* (15) *achieves*

$$\frac{1}{d_1 d_2} \left\|\left\|\widehat{\Theta} - \Theta^*\right\|\right\|_F^2 \leq 48\sqrt{2}\, e^{2\alpha} c_1 \lambda_1 \sqrt{r} \left\|\left\|\widehat{\Theta} - \Theta^*\right\|\right\|_F + 48e^{2\alpha} c_1 \lambda_1 \sum_{j=r+1}^{\min\{d_1, d_2\}} \sigma_j(\Theta^*) \,, \quad (19)$$

*for any* $r \in \{1, \ldots, \min\{d_1, d_2\}\}$ *with probability at least* $1 - 2d^{-3}$ *where* $d = (d_1 + d_2)/2$.

A proof is provided in Appendix D. Optimizing over $r$ gives the following corollaries.

**Corollary 4.1** (**Exact low-rank matrices**). *Suppose* $\Theta^*$ *has rank at most* $r$. *Under the hypotheses of Theorem 3, solving the optimization* (15) *with the choice of the regularization parameter* $\lambda \in [8\lambda_1, c_1\lambda_1]$ *achieves with probability at least* $1 - 2d^{-3}$,

$$\frac{1}{\sqrt{d_1 d_2}} \left\|\left\|\widehat{\Theta} - \Theta^*\right\|\right\|_F \leq 48\sqrt{2}e^{3\alpha} c_1 \sqrt{\frac{r(d_1 + d_2)\log d}{n}} \,. \quad (20)$$

This corollary shows that the number of samples $n$ needs to scale as $O(r(d_1 + d_2)\log d)$ in order to achieve an arbitrarily small error. This is only a logarithmic factor larger than the degrees of freedom. We provide a fundamental lower bound on the error, that matches the upper bound up to a logarithmic factor. For approximately low-rank matrices in an $\ell_1$-ball as defined in (10), we show an upper bound on the error, whose error exponent reduces from one to $(2 - q)/2$.

**Corollary 4.2** (**Approximately low-rank matrices**). *Suppose* $\Theta^* \in \mathbb{B}_q(\rho_q)$ *for some* $q \in (0, 1]$ *and* $\rho_q > 0$. *Under the hypotheses of Theorem 3, solving the optimization* (15) *with the choice of the regularization parameter* $\lambda \in [8\lambda_1, c_1\lambda_1]$ *achieves with probability at least* $1 - 2d^{-3}$,

$$\frac{1}{\sqrt{d_1 d_2}} \left\|\left\|\widehat{\Theta} - \Theta^*\right\|\right\|_F \leq \frac{2\sqrt{\rho_q}}{\sqrt{d_1 d_2}} \left(48\sqrt{2}c_1 e^{3\alpha} \sqrt{\frac{d_1 d_2 (d_1 + d_2)\log d}{n}}\right)^{\frac{2-q}{2}} \,. \quad (21)$$

Since the proof is almost identical to the proof of Corollary 3.2 in Appendix B, we omit it.

**Theorem 4.** *Suppose* $\Theta^*$ *has rank* $r$. *Under the described sampling model, there is a universal constant* $c > 0$ *such that that the minimax rate where the infimum is taken over all measurable functions over the observed purchase history* $\{(u_i, v_i, S_i, T_i)\}_{i \in [n]}$ *is lower bounded by*

$$\inf_{\widehat{\Theta}} \sup_{\Theta^* \in \Omega_\alpha} \mathbb{E}\left[\frac{1}{\sqrt{d_1 d_2}} \left\|\left\|\widehat{\Theta} - \Theta^*\right\|\right\|_F\right] \geq c \min\left\{\sqrt{\frac{e^{-5\alpha} r(d_1 + d_2)}{n}}, \frac{\alpha(d_1 + d_2)}{\sqrt{d_1 d_2 \log d}}\right\} \,. \quad (22)$$

See Appendix E.1 for the proof. The first term is dominant, and when the sample size is comparable to the latent dimension of the problem, Theorem 3 is minimax optimal up to a logarithmic factor.

## 5  Discussion

We presented a convex program to learn MNL parameters from ordinal data, motivated by two scenarios: recommendation systems and bundled purchases. We take the first principle approach of identifying the fundamental limits and also developing efficient algorithms matching those fundamental trade offs. There are several remaining challenges. $(a)$ Nuclear norm minimization, while polynomial-time, is still slow. We want first-order methods that are efficient with provable guarantees. The main challenge is providing a good initialization to start such non-convex approaches. $(b)$ For simpler models, such as the PL model, more general sampling over a graph has been studied. We want analytical results for more general sampling. $(c)$ The practical use of the model and the algorithm needs to be tested on real datasets on purchase history and recommendations.

**Acknowledgments**

This research is supported in part by NSF CMMI award MES-1450848 and NSF SaTC award CNS-1527754.

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
