[Supplementary Material]

# Supplementary material for "Collaboratively Learning Preferences from Ordinal Data"

## A   Proof of Theorem 1

We first introduce some additional notations used in the proof. Recall that $\mathcal{L}(\Theta)$ is the log likelihood function. Let $\nabla \mathcal{L}(\Theta) \in \mathbb{R}^{d_1 \times d_2}$ denote its gradient such that $\nabla_{ij} \mathcal{L}(\Theta) = \frac{\partial \mathcal{L}(\Theta)}{\partial \Theta_{ij}}$. Let $\nabla^2 \mathcal{L}(\Theta) \in \mathbb{R}^{d_1 d_2 \times d_1 d_2}$ denote its Hessian matrix such that $\nabla^2_{ij,i'j'} \mathcal{L}(\Theta) = \frac{\partial^2 \mathcal{L}(\Theta)}{\partial \Theta_{ij} \partial \Theta_{i'j'}}$. By the definition of $\mathcal{L}(\Theta)$ in (4), we have

$$\nabla \mathcal{L}(\Theta^*) \;=\; -\frac{1}{k\,d_1} \sum_{i=1}^{d_1} \sum_{\ell=1}^{k} e_i (e_{v_{i,\ell}} - p_{i,\ell})^T \,, \tag{23}$$

where $p_{i,\ell}$ denotes the conditional choice probability at $\ell$-th position. Precisely, $p_{i,\ell} = \sum_{j \in S_{i,\ell}} p_{j|(i,\ell)} e_j$ where $p_{j|(i,\ell)}$ is the probability that item $j$ is chosen at $\ell$-th position from the top by the user $i$ conditioned on the top $\ell-1$ choices such that $p_{j|(i,\ell)} \equiv \mathbb{P}\{v_{i,\ell} = j | v_{i,1}, \ldots, v_{i,\ell-1}, S_i\} = e^{\Theta^*_{ij}}/(\sum_{j' \in S_{i,\ell}} e^{\Theta_{ij'}})$ and $S_{i,\ell} \equiv S_i \setminus \{v_{i,1}, \ldots, v_{i,\ell-1}\}$, where $S_i$ is the set of alternatives presented to the $i$-th user and $v_{i,\ell}$ is the item ranked at the $\ell$-th position by the user $i$. Notice that for $i \neq i'$, $\frac{\partial^2 \mathcal{L}(\Theta)}{\partial \Theta_{ij} \partial \Theta_{i'j'}} = 0$ and the Hessian is

$$\frac{\partial^2 \mathcal{L}(\Theta)}{\partial \Theta_{ij} \partial \Theta_{ij'}} \;=\; \frac{1}{k\,d_1} \sum_{\ell=1}^{k} \mathbb{I}(j \in S_{i,\ell}) \frac{\partial p_{j|(i,\ell)}}{\partial \Theta_{ij'}}$$

$$=\; \frac{1}{k\,d_1} \sum_{\ell=1}^{k} \mathbb{I}(j, j' \in S_{i,\ell}) \left( p_{j|(i,\ell)} \mathbb{I}(j = j') - p_{j|(i,\ell)} p_{j'|(i,\ell)} \right) . \tag{24}$$

This Hessian matrix is a block-diagonal matrix $\nabla^2 \mathcal{L}(\Theta) = \mathrm{diag}(H^{(1)}(\Theta), \ldots, H^{(d_1)}(\Theta))$ with

$$H^{(i)}(\Theta) = \frac{1}{k\,d_1} \sum_{\ell=1}^{k} \left( \mathrm{diag}(p_{i,\ell}) - p_{i,\ell} p_{i,\ell}^T \right) . \tag{25}$$

Let $\Delta = \Theta^* - \widehat{\Theta}$ where $\widehat{\Theta}$ is the optimal solution of the convex program in (3). We first introduce three key technical lemmas. The first lemma follows from Lemma 1 of [22], and shows that $\Delta$ is approximately low-rank.

**Lemma A.1.** *If $\lambda \geq 2 \|\|\nabla \mathcal{L}(\Theta^*)\|\|_2$, then we have*

$$\|\|\Delta\|\|_{\mathrm{nuc}} \;\leq\; 4\sqrt{2r} \|\|\Delta\|\|_{\mathrm{F}} + 4 \sum_{j=\rho+1}^{\min\{d_1,d_2\}} \sigma_j(\Theta^*) \,, \tag{26}$$

*for all $\rho \in [\min\{d_1, d_2\}]$.*

The following lemma provides a bound on the gradient using the concentration of measure for sum of independent random matrices [25].

**Lemma A.2.** *For any positive constant $c \geq 1$ and $k \leq (1/e)\, d_2(4 \log d_2 + \log d_1)$, with probability at least $1 - 2d^{-c} - d_2^{-3}$,*

$$\|\|\nabla \mathcal{L}(\Theta^*)\|\|_2 \leq \sqrt{\frac{4(1+c)\log d}{k\,d_1^2}} \max\left\{ \sqrt{d_1/d_2}, \; e^{2\alpha} \sqrt{4(1+c)\log(d)}(8\log d_2 + 2\log d_1)\log k \right\} . \tag{27}$$

Since we are typically interested in the regime where the number of samples is much smaller than the dimension $d_1 \times d_2$ of the problem, the Hessian is typically not positive definite. However, when we restrict our attention to the vectorized $\Delta$ with relatively small nuclear norm, then we can prove restricted strong convexity, which gives the following bound.

**Lemma A.3** (**Restricted Strong Convexity for collaborative ranking**). *Fix any $\Theta \in \Omega_\alpha$ and assume $24 \leq k \leq \min\{d_1^2, (d_1^2 + d_2^2)/(2d_1)\} \log d$. Under the random sampling model of the alternatives $\{j_{i\ell}\}_{i \in [d_1], \ell \in [k]}$ and the random outcome of the comparisons described in section 1, with probability larger than $1 - 2d^{-2^{18}}$,*

$$\mathrm{Vec}(\Delta)^T \, \nabla^2 \mathcal{L}(\Theta) \, \mathrm{Vec}(\Delta) \;\; \geq \;\; \frac{e^{-4\alpha}}{24 \, d_1 d_2} \|\Delta\|_{\mathrm{F}}^2 \, , \tag{28}$$

*for all $\Delta$ in $\mathcal{A}$ where*

$$\mathcal{A} = \left\{ \Delta \in \mathbb{R}^{d_1 \times d_2} \, \big| \, \|\Delta\|_\infty \leq 2\alpha \, , \, \sum_{j \in [d_2]} \Delta_{ij} = 0 \, \textit{for all } i \in [d_1] \textit{ and } \|\Delta\|_{\mathrm{F}}^2 \geq \mu \|\Delta\|_{\mathrm{nuc}} \right\} . \tag{29}$$

*with*

$$\mu \;\; \equiv \;\; 2^{10} \, e^{2\alpha} \, \alpha \, d_2 \sqrt{\frac{d_1 \log d}{k \, \min\{d_1, d_2\}}} \, . \tag{30}$$

Building on these lemmas, the proof of Theorem 1 is divided into the following two cases. In both cases, we will show that

$$\|\Delta\|_{\mathrm{F}}^2 \;\; \leq \;\; 72 \, e^{4\alpha} c_0 \lambda_0 \, d_1 d_2 \, \|\Delta\|_{\mathrm{nuc}} \, , \tag{31}$$

with high probability. Applying Lemma A.1 proves the desired theorem. We are left to show Eq. (31) holds.

**Case 1: Suppose $\|\Delta\|_{\mathrm{F}}^2 \geq \mu \|\Delta\|_{\mathrm{nuc}}$.** With $\Delta = \Theta^* - \widehat{\Theta}$, the Taylor expansion yields

$$\mathcal{L}(\widehat{\Theta}) = \mathcal{L}(\Theta^*) - \langle\!\langle \nabla \mathcal{L}(\Theta^*), \Delta \rangle\!\rangle + \frac{1}{2} \mathrm{Vec}(\Delta) \nabla^2 \mathcal{L}(\Theta) \mathrm{Vec}^T(\Delta), \tag{32}$$

where $\Theta = a\widehat{\Theta} + (1-a)\Theta^*$ for some $a \in [0, 1]$. It follows from Lemma A.3 that with probability at least $1 - 2d^{-2^{18}}$,

$$\mathcal{L}(\widehat{\Theta}) - \mathcal{L}(\Theta^*) \;\; \geq \;\; -\langle\!\langle \nabla \mathcal{L}(\Theta^*), \Delta \rangle\!\rangle + \frac{e^{-4\alpha}}{48 \, d_1 d_2} \|\Delta\|_{\mathrm{F}}^2$$

$$\geq \;\; -\|\nabla\mathcal{L}(\Theta^*)\|_2 \|\Delta\|_{\mathrm{nuc}} + \frac{e^{-4\alpha}}{48 \, d_1 d_2} \|\Delta\|_{\mathrm{F}}^2 \, .$$

From the definition of $\widehat{\Theta}$ as an optimal solution of the minimization, we have

$$\mathcal{L}(\widehat{\Theta}) - \mathcal{L}(\Theta^*) \;\; \leq \;\; \lambda \left( \|\Theta^*\|_{\mathrm{nuc}} - \left\|\left\|\widehat{\Theta}\right\|\right\|_{\mathrm{nuc}} \right) \;\; \leq \;\; \lambda \|\Delta\|_{\mathrm{nuc}} \, .$$

By the assumption, we choose $\lambda \geq 480\lambda_0$. In view of Lemma A.2, this implies that $\lambda \geq 2\|\nabla\mathcal{L}(\Theta^*)\|_2$ with probability at least $1 - 2d^{-3}$. It follows that with probability at least $1 - 2d^{-3} - 2d^{-2^{18}}$,

$$\frac{e^{-4\alpha}}{48 d_1 d_2} \|\Delta\|_{\mathrm{F}}^2 \;\; \leq \;\; \left( \lambda + \|\nabla\mathcal{L}(\Theta^*)\|_2 \right) \|\Delta\|_{\mathrm{nuc}} \;\; \leq \;\; \frac{3\lambda}{2} \|\Delta\|_{\mathrm{nuc}} \, .$$

By our assumption on $\lambda \leq c_0 \lambda_0$, this proves the desired bound in Eq. (31)

**Case 2: Suppose $\|\Delta\|_{\mathrm{F}}^2 \leq \mu \|\Delta\|_{\mathrm{nuc}}$.** By the definition of $\mu$ and the fact that $c_0 \geq 480$, it follows that $\mu \leq 72 \, e^{4\alpha} c_0 \lambda_0 \, d_1 d_2$, and we get the same bound as in Eq. (31).

## A.1 Proof of Lemma A.1

Denote the singular value decomposition of $\Theta^*$ by $\Theta^* = U\Sigma V^T$, where $U \in \mathbb{R}^{d_1 \times d_1}$ and $V \in \mathbb{R}^{d_2 \times d_2}$ are orthogonal matrices. For a given $r \in [\min\{d_1, d_2\}]$, Let $U_r = [u_1, \ldots, u_r]$ and $V_r = [v_1, \ldots, v_r]$, where $u_i \in \mathbb{R}^{d_1 \times 1}$ and $v_i \in \mathbb{R}^{d_2 \times 1}$ are the left and right singular vectors corresponding to the $i$-th largest singular value, respectively. Define $T$ to be the subspace spanned

by all matrices in $\mathbb{R}^{d_1 \times d_2}$ of the form $U_r A^T$ or $B V_r^T$ for any $A \in \mathbb{R}^{d_2 \times r}$ or $B \in \mathbb{R}^{d_1 \times r}$, respectively. The orthogonal projection of any matrix $M \in \mathbb{R}^{d_1 \times d_2}$ onto the space $T$ is given by $\mathcal{P}_T(M) = U_r U_r^T M + M V_r V_r^T - U_r U_r^T M V_r V_r^T$. The projection of $M$ onto the complement space $T^\perp$ is $\mathcal{P}_{T^\perp}(M) = (I - U_r U_r^T) M (I - V_r V_r^T)$. The subspace $T$ and the respective projections onto $T$ and $T^\perp$ play crucial a role in the analysis of nuclear norm minimization, since they define the sub-gradient of the nuclear norm at $\Theta^*$. We refer to [21] for more detailed treatment of this topic.

Let $\Delta' = \mathcal{P}_T(\Delta)$ and $\Delta'' = \mathcal{P}_{T^\perp}(\Delta)$. Notice that $\mathcal{P}_T(\Theta^*) = U_r \Sigma_r V_r^T$, where $\Sigma_r \in \mathbb{R}^{r \times r}$ is the diagonal matrix formed by the top $r$ singular values. Since $\mathcal{P}_T(\Theta^*)$ and $\Delta''$ have row and column spaces that are orthogonal, it follows from Lemma 2.3 in [20] that

$$\|\mathcal{P}_T(\Theta^*) - \Delta''\|_{\mathrm{nuc}} = \|\mathcal{P}_T(\Theta^*)\|_{\mathrm{nuc}} + \|\Delta''\|_{\mathrm{nuc}} .$$

Hence, in view of the triangle inequality,

$$
\begin{aligned}
\left\|\widehat{\Theta}\right\|_{\mathrm{nuc}} &= \|\mathcal{P}_T(\Theta^*) + \mathcal{P}_{T^\perp}(\Theta^*) - \Delta' - \Delta''\|_{\mathrm{nuc}} \\
&\geq \|\mathcal{P}_T(\Theta^*) - \Delta''\|_{\mathrm{nuc}} - \|\mathcal{P}_{T^\perp}(\Theta^*) - \Delta'\|_{\mathrm{nuc}} \\
&= \|\mathcal{P}_T(\Theta^*)\|_{\mathrm{nuc}} + \|\Delta''\|_{\mathrm{nuc}} - \|\mathcal{P}_{T^\perp}(\Theta^*) - \Delta'\|_{\mathrm{nuc}} \\
&\geq \|\mathcal{P}_T(\Theta^*)\|_{\mathrm{nuc}} + \|\Delta''\|_{\mathrm{nuc}} - \|\mathcal{P}_{T^\perp}(\Theta^*)\|_{\mathrm{nuc}} - \|\Delta'\|_{\mathrm{nuc}} \\
&= \|\Theta^*\|_{\mathrm{nuc}} + \|\Delta''\|_{\mathrm{nuc}} - 2\|\mathcal{P}_{T^\perp}(\Theta^*)\|_{\mathrm{nuc}} - \|\Delta'\|_{\mathrm{nuc}}. \quad (33)
\end{aligned}
$$

Because $\widehat{\Theta}$ is an optimal solution, we have

$$\lambda \left( \left\|\widehat{\Theta}\right\|_{\mathrm{nuc}} - \|\Theta^*\|_{\mathrm{nuc}} \right) \leq \mathcal{L}(\Theta^*) - \mathcal{L}(\widehat{\Theta}) \overset{(a)}{\leq} \langle\!\langle \Delta, \nabla\mathcal{L}(\Theta^*) \rangle\!\rangle \overset{(b)}{\leq} \|\Delta\|_{\mathrm{nuc}} \|\nabla\mathcal{L}(\Theta^*)\|_2 \leq \frac{\lambda}{2}\|\Delta\|_{\mathrm{nuc}},$$
$$(34)$$

where $(a)$ holds due to the convexity of $\mathcal{L}$; $(b)$ follows from the Cauchy-Schwarz inequality; the last inequality holds due to the assumption that $\lambda \geq 2\|\nabla\mathcal{L}(\Theta^*)\|_2$. Combining (33) and (34) yields

$$2 \left( \|\Delta''\|_{\mathrm{nuc}} - 2\|\mathcal{P}_{T^\perp}(\Theta^*)\|_{\mathrm{nuc}} - \|\Delta'\|_{\mathrm{nuc}} \right) \leq \|\Delta\|_{\mathrm{nuc}} \leq \|\Delta'\|_{\mathrm{nuc}} + \|\Delta''\|_{\mathrm{nuc}}.$$

Thus $\|\Delta''\|_{\mathrm{nuc}} \leq 3\|\Delta'\|_{\mathrm{nuc}} + 4\|\mathcal{P}_{T^\perp}(\Theta^*)\|_{\mathrm{nuc}}$. By triangle inequality,

$$\|\Delta\|_{\mathrm{nuc}} \leq 4\|\Delta'\|_{\mathrm{nuc}} + 4\|\mathcal{P}_{T^\perp}(\Theta^*)\|_{\mathrm{nuc}} .$$

Notice that $\Delta' = U_r U_r^T \Delta + (I - U_r U_r^T)\Delta V_r V_r^T$. Both $U_r U_r^T \Delta$ and $(I - U_r U_r^T)\Delta V_r V_r^T$ have rank at most $r$. Thus $\Delta'$ has rank at most $2r$. Hence, $\|\Delta'\|_{\mathrm{nuc}} \leq \sqrt{2r}\|\Delta'\|_{\mathrm{F}} \leq \sqrt{2r}\|\Delta\|_{\mathrm{F}}$. Then the theorem follows because $\|\mathcal{P}_{T^\perp}(\Theta^*)\|_{\mathrm{nuc}} = \sum_{j=r+1}^{\min\{d_1, d_2\}} \sigma_j(\Theta^*)$.

## A.2 Proof of Lemma A.2

Define $X_i = -e_i \sum_{\ell=1}^k (e_{v_{i,\ell}} - p_{i,\ell})^T$ such that $\nabla\mathcal{L}(\Theta^*) = \frac{1}{k\,d_1} \sum_{i=1}^{d_1} X_i$, which is a sum of $d_1$ independent random matrices. Although $\|X_i\|_2$ can be as large as $O(k)$, this occurs with very low probability. We make this precise in the following lemma and focus on the case where $\|X_i\|_2 = O(\sqrt{k})$ for all $i \in [d_1]$.

**Lemma A.4.** *For a fixed $i \in [d_1]$ and $j \in [d_2]$, if $k \leq (1/e)\, d_2\, (4\log d_2 + \log d_1)$, then the number of times the item $j$ is observed by the user $i$ is at most $8(\log d_2) + 2(\log d_1)$ with probability larger than $1 - 1/(d_2^4 d_1)$.*

Proof is given in the end of this Section. Applying union bound over the $d_1$ items and $d_2$ users, we have the multiplicity in sampling for any item for all users is bounded by $8(\log d_2) + 2(\log d_1)$ with probability at least $1 - d_2^{-3}$. We denote this event by $\mathcal{A}$ and let $\mathbb{I}(\mathcal{A})$ be the indicator function that all the multiplicities in sampling are bounded. We first upper bound $\|(\sum_i X_i)\mathbb{I}(\mathcal{A})\|_2$ using the

Matrix Bernstein inequality [25].

$$\left\|\|X_i \mathbb{I}(\mathcal{A})\|\right\|_2 = \left\|\mathbb{I}(\mathcal{A}) \sum_{\ell=1}^{k} \left(e_{v_{i,\ell}} - p_{i,\ell}\right)\right\|$$

$$\overset{(a)}{\leq} \left\|\mathbb{I}(\mathcal{A}) \sum_{\ell=1}^{k} e_{v_{i,l}}\right\| + \left\|\mathbb{I}(\mathcal{A}) \sum_{\ell=1}^{k} p_{i,\ell}\right\|$$

$$\overset{(b)}{\leq} (8(\log d_2) + 2(\log d_1))\sqrt{\min\{k, d_2\}}\left(1 + \left(\sum_{\ell=1}^{k} \frac{e^{2\alpha}}{\ell}\right)\right)$$

$$\overset{(c)}{\leq} \sqrt{k}(8(\log d_2) + 2(\log d_1))\left(1 + 2e^{2\alpha}\ \log k\right)$$

$$\leq 3\sqrt{k}(8(\log d_2) + 2(\log d_1))e^{2\alpha}\ \log k\ , \tag{35}$$

where $(a)$ is by triangle inequality, $(b)$ is because under the given event $\mathcal{A}$ each term in $\sum_{\ell} e_{v_{i,\ell}}$ and $\sum_{l} p_{i,\ell}$ are upper bounded by $\log d_2$ and $\left(\sum_{\ell=1}^{k} \frac{e^{2\alpha}}{\ell}\right) \log d_2$ respectively and because there can be at most $\min\{\sqrt{d_2}, k\}$ non-zero entries in the two vectors $\sum_{\ell} e_{v_{i,\ell}}$ and $\sum_{\ell} p_{i,\ell}$ and, $(c)$ is due to the fact that $k$-th harmonic number $\sum_{\ell=1}^{k} \frac{1}{\ell}$ is upper bounded by $\log k$. We also have,

$$\left\|\left\|\sum_i \mathbb{E}\left[X_i X_i^T \mathbb{I}(\mathcal{A})\right]\right\|\right\|_2 \leq \left\|\left\|\sum_i \mathbb{E}\left[X_i X_i^T\right]\right\|\right\|_2 \leq \left\|\left\|\sum_{i=1}^{d_1} e_i e_i^T \mathbb{E}\left[\sum_{\ell,\ell'=1}^{k} \left(e_{v_{i,\ell}} - p_{i,\ell}\right)^T \left(e_{v_{i,\ell'}} - p_{i,\ell'}\right)\right]\right\|\right\|_2$$

$$= \left\|\left\|\sum_{i=1}^{d_1} e_i e_i^T \mathbb{E}\left[\sum_{\ell=1}^{k} \left(e_{v_{i,\ell}} - p_{i,\ell}\right)^T \left(e_{v_{i,\ell}} - p_{i,\ell}\right)\right]\right\|\right\|_2$$

$$= \left\|\left\|\sum_{i=1}^{d_1} e_i e_i^T \mathbb{E}\left[\sum_{\ell=1}^{k} e_{v_{i,\ell}}^T e_{v_{i,\ell}} - p_{i,\ell}^T p_{i,\ell}\right]\right\|\right\|_2$$

$$\leq \left\|\left\|\sum_{i=1}^{d_1} e_i e_i^T \mathbb{E}\left[\sum_{\ell=1}^{k} e_{v_{i,\ell}}^T e_{v_{i,\ell}}\right]\right\|\right\|_2$$

$$= k\left\|\|\mathbf{I}_{d_1 \times d_1}\|\right\|_2 = k, \tag{36}$$

and

$$\left\|\left\|\sum_{i=1}^{d_1} \mathbb{E}\left[X_i^T X_i \mathbb{I}(\mathcal{A})\right]\right\|\right\|_2 \leq \left\|\left\|\sum_{i=1}^{d_1} \mathbb{E}\left[X_i^T X_i\right]\right\|\right\|_2$$

$$\leq \left\|\left\|\sum_{i=1}^{d_1} \mathbb{E}\left[\sum_{\ell,\ell'=1}^{k} (e_{v_{i,\ell}} - p_{i,\ell})(e_{v_{i,\ell'}} - p_{i,\ell'})^T\right]\right\|\right\|_2$$

$$= \left\|\left\|\sum_{i=1}^{d_1} \mathbb{E}\left[\sum_{\ell=1}^{k} (e_{v_{i,\ell}} - p_{i,\ell})(e_{v_{i,\ell}} - p_{i,\ell})^T\right]\right\|\right\|_2 \tag{37}$$

$$= \left\|\left\|\sum_{i=1}^{d_1} \mathbb{E}\left[\sum_{\ell=1}^{k} e_{v_{i,\ell}} e_{v_{i,\ell}}^T - p_{i,\ell} p_{i,\ell}^T\right]\right\|\right\|_2$$

$$\leq \left\|\left\|\sum_{i=1}^{d_1} \mathbb{E}\left[\sum_{\ell=1}^{k} e_{v_{i,\ell}} e_{v_{i,\ell}}^T\right]\right\|\right\|_2$$

$$= \left\|\left\|\sum_{i=1}^{d_1} \frac{k}{d_2} \mathbf{I}_{d_2 \times d_2}\right\|\right\|_2 = \frac{kd_1}{d_2}\ . \tag{38}$$

By matrix Bernstein inequality [25],

$$\mathbb{P}\left(\|\nabla\mathcal{L}(\Theta^*)\mathbb{I}(\mathcal{A})\|_2 > t\right) \leq (d_1 + d_2)\exp\left(\frac{-k^2\,d_1^2\,t^2/2}{(d_1 k/\min\{d_2, d_1\}) + (3e^{2\alpha} k^{3/2} d_1(8(\log d_2) + 2(\log d_1))\log k\ t/3)}\right),$$

which gives the tail probability of $2d^{-c}$ for the choice of

$$
\begin{aligned}
t &= \max\left\{\sqrt{\frac{4(1+c)\,\log d}{k\,d_1\,\min\{d_2,d_1\}}}\,,\ \frac{4(1+c)e^{2\alpha}\log(d)\,(8(\log d_2)+2(\log d_1))\log k}{k^{1/2}\,d_1}\right\} \\
&= \frac{\sqrt{4(1+c)\,\log d}}{k^{1/2}\,d_1}\max\left\{\sqrt{d_1/d_2}\,,\ e^{2\alpha}\sqrt{4(1+c)\log(d)}\,(8(\log d_2)+2(\log d_1))\log k\right\}.
\end{aligned}
$$

Now with a high probability of $1-\frac{2}{d^c}-\frac{1}{d_2^3}$ the desired bound is true.

## A.3 Proof of Lemma A.2

In a classical balls-in-bins setting, we consider $k$ as the number of balls and $d_2$ as the number of bins. We can consider the number of balls in a particular bin as the number of times the user $i$ observes item $j$. Let the event that this number is at least $\delta$ be denoted by the event $A_\delta^j$. Then, $\mathbb{P}\left\{A_\delta^j\right\} \leq \binom{k}{\delta}\frac{1}{d_2^\delta} \leq \left(\frac{ke}{d_2\delta}\right)^\delta$. Using the fact that $(1/x)^x \leq a$ for any $x \geq (2\log(1/a))/(\log\log(1/a))$, we let $x = d_2\delta/(ke)$ to get

$$
\left(\frac{ke}{d_2\delta}\right)^\delta \leq a^{\frac{ke}{d_2}}\,,
$$

for $\delta \geq (ke/d_2)(2\log(1/a))/(\log\log(1/a))$. Choosing $a = (1/d_2^4 d_1)^{d_2/ke}$, we have $\mathbb{P}\left\{A_\delta^j\right\} \leq 1/(d_1 d_2^4)$, for a choice of $\delta = 2\,\log(d_2^4 d_1) \geq 2\log(d_2^4 d_1)/(\log((d_2/ke)\log(d_2^4 d_1)))$.

## A.4 Proof of Lemma A.3

Recall that the Hessian matrix is a block-diagonal matrix with the $i$-th block $H^{(i)}(\Theta)$ given by (25). We use the following remark from [12] to bound the Hessian.

**Remark A.5.** *[12, Claim 1] Given* $\theta \in \mathbb{R}^r$, *let* $p$ *be the column probability vector with* $p_i = e^{\theta_i}/(e^{\theta_1}+\cdots+e^{\theta_\rho})$ *for each* $i \in [\rho]$ *and for any positive integer* $\rho$. *If* $|\theta_i| \leq \alpha$, *for all* $i \in [\rho]$, *then*

$$
e^{2\alpha}\left(\mathrm{diag}(p) - pp^T\right) \succeq \frac{1}{\rho}\mathrm{diag}(\mathbb{1}) - \frac{1}{\rho^2}\mathbb{1}\mathbb{1}^T\,.
$$

By letting $\mathbb{1}_{S_{i,\ell}} = \sum_{j\in S_{i,\ell}} e_j$ and applying the above claim, we have

$$
\begin{aligned}
e^{2\alpha}H^{(i)}(\Theta) &\succeq \frac{1}{k\,d_1}\sum_{\ell=1}^{k}\left(\frac{1}{k-\ell+1}\mathrm{diag}(\mathbb{1}_{S_{i,\ell}}) - \frac{1}{(k-\ell+1)^2}\mathbb{1}_{S_{i,\ell}}\mathbb{1}_{S_{i,\ell}}^T\right) \\
&= \frac{1}{2\,k\,d_1}\sum_{\ell=1}^{k}\frac{1}{(k-\ell+1)^2}\sum_{j,j'\in S_{i,\ell}}(e_j - e_{j'})(e_j - e_{j'})^T \\
&\succeq \frac{1}{2\,k^3\,d_1}\sum_{\ell=1}^{k}\sum_{j,j'\in S_{i,\ell}}(e_j - e_{j'})(e_j - e_{j'})^T.
\end{aligned}
$$

Hence,

$$
\begin{aligned}
\mathrm{Vec}(\Delta)\nabla^2\mathcal{L}(\Theta)\mathrm{Vec}^T(\Delta) &= \sum_{i=1}^{d_1}(\Delta^T e_i)^T H^{(i)}(\Theta)(\Delta^T e_i) \\
&\geq \frac{e^{-2\alpha}}{2\,k^3\,d_1}\sum_{i=1}^{d_1}\sum_{\ell=1}^{k}\sum_{j,j'\in S_{i,\ell}}\left\|\left|e_i^T\Delta(e_j - e_{j'})\right|\right\|_2^2.
\end{aligned}
$$

By changing the order of the summation, we get that

$$\sum_{\ell=1}^{k} \sum_{j,j' \in S_{i,\ell}} \||e_i^T \Delta (e_j - e_{j'})\||_2^2 = \sum_{\ell,\ell'=1}^{k} \langle\!\langle \Delta, e_{i,j_{i,\ell}} - e_{i,j_{i,\ell'}} \rangle\!\rangle^2 \sum_{\ell''=1}^{k} \mathbb{I}\big( \sigma_i(j_{i,\ell''}) \leq \min\{\sigma_i(j_{i,\ell}), \sigma_i(j_{i,\ell'})\} \big).$$

Define

$$\chi_{i,\ell,\ell',\ell''} \equiv \mathbb{I}\big( \sigma_i(j_{i,\ell''}) \leq \min\{\sigma_i(j_{i,\ell}), \sigma_i(j_{i,\ell'})\} \big), \tag{39}$$

and let

$$H(\Delta) \equiv \frac{e^{-2\alpha}}{2\,k^3\,d_1} \sum_{i=1}^{d_1} \sum_{\ell,\ell'=1}^{k} \langle\!\langle \Delta, e_{i,j_{i,\ell}} - e_{i,j_{i,\ell'}} \rangle\!\rangle^2 \sum_{\ell''=1}^{k} \chi_{i,\ell,\ell',\ell''}.$$

Then we have $\mathrm{Vec}^T(\Delta)\nabla^2 \mathcal{L}(\Theta)\mathrm{Vec}(\Delta) \geq H(\Delta)$. To prove the theorem, it suffices to bound $H(\Delta)$ from the below. First, we prove a lower bound on the expectation $\mathbb{E}[H(\Delta)]$. Notice that for $\ell \neq \ell'$, the conditional expectation of $\chi_{i,\ell,\ell',\ell''}$'s, given the set of alternatives presented to user $i$ is

$$\mathbb{E}\Big[ \sum_{\ell''=1}^{k} \chi_{i,\ell,\ell',\ell''} \,\big|\, j_{i,1}, \ldots, j_{i,k} \Big] = 1 + \sum_{\ell'' \neq \ell,\ell'} \frac{\exp(\theta_{i,j_{i,\ell''}})}{\exp(\theta_{i,j_{i,\ell''}}) + \exp(\theta_{i,j_{i,\ell'}}) + \exp(\theta_{i,j_{i,\ell}})}$$

$$\geq 1 + \frac{k-2}{1+2e^{2\alpha}} \geq \frac{k}{3e^{2\alpha}}.$$

Then,

$$\begin{aligned}
\mathbb{E}[H(\Delta)] &= \frac{e^{-2\alpha}}{2\,k^3\,d_1} \sum_{i,\ell,\ell'} \mathbb{E}\Big[ \langle\!\langle \Delta, e_{i,j_{i,\ell}} - e_{i,j_{i,\ell'}} \rangle\!\rangle^2 \mathbb{E}\Big[ \sum_{\ell''=1}^{k} \chi_{i,\ell,\ell',\ell''} \,\big|\, j_{i,1}, \ldots, j_{i,k} \Big] \Big] \\
&\geq \frac{e^{-4\alpha}}{6\,k^2\,d_1} \sum_{i=1}^{d_1} \sum_{\ell,\ell' \in [k]} \mathbb{E}\Big[ \langle\!\langle \Delta, e_{i,j_{i,\ell}} - e_{i,j_{i,\ell'}} \rangle\!\rangle^2 \Big] \\
&= \frac{e^{-4\alpha}}{6\,k^2\,d_1} \sum_{i=1}^{d_1} \sum_{\ell \neq \ell' \in [k]} \left( \frac{2}{d_2} \sum_{j=1}^{d_2} \Delta_{ij}^2 - \frac{2}{d_2^2} \sum_{j,j'=1}^{d_2} \Delta_{ij} \Delta_{ij'} \right) \\
&= \frac{e^{-4\alpha}(k-1)}{3\,k\,d_1\,d_2} \||\Delta\||_{\mathrm{F}}^2 , \tag{40}
\end{aligned}$$

where the last equality holds because $\sum_{j \in [d_2]} \Delta_{ij} = 0$ for $\Delta \in \Omega_{2\alpha}$ and for all $i \in [d_1]$.

We are left to prove that $H(\Delta)$ cannot deviate from its mean too much. Suppose there exists a $\Delta \in \mathcal{A}$ such that Eq. (28) is violated, i.e. $H(\Delta) < (e^{-4\alpha}/(24\,d_1 d_2))\||\Delta\||_{\mathrm{F}}^2$. We will show this happens with a small probability. From Eq. (40), we get that for $k \geq 24$,

$$\begin{aligned}
\mathbb{E}[H(\Delta)] - H(\Delta) &\geq \frac{(7k-8)}{24k} \frac{e^{-4\alpha}}{d_1\,d_2} \||\Delta\||_{\mathrm{F}}^2 \\
&\geq \frac{(20/3)\,e^{-4\alpha}}{24\,d_1 d_2} \||\Delta\||_{\mathrm{F}}^2 . \tag{41}
\end{aligned}$$

We use a peeling argument as in [22, Lemma 3], [26] to upper bound the probability that Eq. (41) is true. We first construct the following family of subsets to cover $\mathcal{A}$ such that $\mathcal{A} \subseteq \bigcup_{\ell=1}^{\infty} \mathcal{S}_\ell$. Recall $\mu = 2^{10} e^{2\alpha} \alpha d_2 \sqrt{(d_1 \log d)/(k \min\{d_1, d_2\})}$, define in (30). Notice that since for any $\Delta \in \mathcal{A}$, $\||\Delta\||_{\mathrm{F}}^2 \geq \mu \||\Delta\||_{\mathrm{nuc}} \geq \mu \||\Delta\||_{\mathrm{F}}$, it follows that $\||\Delta\||_{\mathrm{F}} \geq \mu$. Then, we can cover $\mathcal{A}$ with the family of sets

$$\mathcal{S}_\ell = \Big\{ \Delta \in \mathbb{R}^{d_1 \times d_2} \,\Big|\, \||\Delta\||_{\infty} \leq 2\alpha, \ \beta^{\ell-1}\mu \leq \||\Delta\||_{\mathrm{F}} \leq \beta^\ell \mu, \ \sum_{j \in [d_2]} \Delta_{ij} = 0 \text{ for all } i \in [d_1], \text{ and } \||\Delta\||_{\mathrm{nuc}} \leq \beta^{2\ell}\mu \Big\},$$

where $\beta = \sqrt{10/9}$ and for $\ell \in \{1, 2, 3, \ldots\}$. This implies that when there exists a $\Delta \in \mathcal{A}$ such that (41) holds, then there exists an $\ell \in \mathbb{Z}_+$ such that $\Delta \in \mathcal{S}_\ell$ and

$$
\begin{aligned}
\mathbb{E}[H(\Delta)] - H(\Delta) &\geq \frac{(20/3)\, e^{-4\alpha}}{24\, d_1 d_2} \beta^{2(\ell-1)} \mu^2 \\
&\geq \frac{e^{-4\alpha}}{4\, d_1 d_2} \beta^{2\ell} \mu^2 \,.
\end{aligned}
\tag{42}
$$

Applying the union bound over $\ell \in \mathbb{Z}_+$, we get from (41) and (42) that

$$
\begin{aligned}
\mathbb{P}\left\{ \exists \Delta \in \mathcal{A} \,,\ H(\Delta) < \frac{e^{-4\alpha}}{24\, d_1 d_2} \|\!|\Delta|\!\|_{\mathrm{F}}^2 \right\} &\leq \sum_{\ell=1}^{\infty} \mathbb{P}\left\{ \sup_{\Delta \in \mathcal{S}_\ell} \big( \mathbb{E}[H(\Delta)] - H(\Delta) \big) > \frac{e^{-4\alpha}}{4\, d_1 d_2} (\beta^\ell \mu)^2 \right\} \\
&\leq \sum_{\ell=1}^{\infty} \mathbb{P}\left\{ \sup_{\Delta \in \mathcal{B}(\beta^\ell \mu)} \big( \mathbb{E}[H(\Delta)] - H(\Delta) \big) > \frac{e^{-4\alpha}}{4\, d_1 d_2} (\beta^\ell \mu)^2 \right\}, \quad (43)
\end{aligned}
$$

where we define a new set $\mathcal{B}(D)$ such that $\mathcal{S}_\ell \subseteq \mathcal{B}(\beta^\ell \mu)$:

$$
\mathcal{B}(D) = \Big\{ \Delta \in \mathbb{R}^{d_1 \times d_2} \,\big|\, \|\Delta\|_\infty \leq 2\alpha, \|\!|\Delta|\!\|_{\mathrm{F}} \leq D, \sum_{j \in [d_2]} \Delta_{ij} = 0 \text{ for all } i \in [d_1], \mu\|\!|\Delta|\!\|_{\mathrm{nuc}} \leq D^2 \Big\} \,.
\tag{44}
$$

The following key lemma provides the upper bound on this probability.

**Lemma A.6.** *For* $(16 \min\{d_1, d_2\} \log d)/(3d_1) \leq k \leq d_1^2 \log d$,

$$
\mathbb{P}\left\{ \sup_{\Delta \in \mathcal{B}(D)} \big( \mathbb{E}[H(\Delta)] - H(\Delta) \big) \geq \frac{e^{-4\alpha}}{4 d_1 d_2} D^2 \right\} \leq \exp\left\{ -\frac{e^{-4\alpha}\, k\, D^4}{2^{19} \alpha^4 d_1 d_2^2} \right\}.
\tag{45}
$$

Let $\eta = \exp\left( -\frac{e^{-4\alpha} 4k(\beta - 1.002)\mu^4}{2^{19} \alpha^4 d_1 d_2^2} \right)$. Applying the tail bound to (43), we get

$$
\begin{aligned}
\mathbb{P}\left\{ \exists \Delta \in \mathcal{A} \,,\ H(\Delta) < \frac{e^{-4\alpha}}{24\, d_1 d_2} \|\!|\Delta|\!\|_{\mathrm{F}}^2 \right\} &\leq \sum_{\ell=1}^{\infty} \exp\left\{ -\frac{e^{-4\alpha} k(\beta^\ell \mu)^4}{2^{19} \alpha^4 d_1 d_2^2} \right\} \\
&\overset{(a)}{\leq} \sum_{\ell=1}^{\infty} \exp\left\{ -\frac{e^{-4\alpha} 4k\ell(\beta - 1.002)\mu^4}{2^{19} \alpha^4 d_1 d_2^2} \right\} \\
&\leq \frac{\eta}{1 - \eta},
\end{aligned}
$$

where $(a)$ holds because $\beta^x \geq x \log \beta \geq x(\beta - 1.002)$ for the choice of $\beta = \sqrt{10/9}$. By the definition of $\mu$,

$$
\eta = \exp\left\{ -\frac{2^{23}\, e^{4\alpha} d_2^2 d_1 (\log d)^2 (\beta - 1.002)}{k(\min\{d_1, d_2\})^2} \right\} \leq \exp\{-2^{18} \log d\} \,,
$$

where the last inequality follows from the assumption that $k \leq \max\{d_1, d_2^2/d_1\} \log d = (d_2^2 d_1 \log d)/(\min\{d_1, d_2\})^2$, and $\beta - 1.002 \geq 2^{-5}$. Since for $d \geq 2$, $\exp\{-2^{18} \log d\} \leq 1/2$ and thus $\eta \leq 1/2$, the lemma follows by assembling the last two displayed inequalities.

## A.5 Proof of Lemma A.6

Recall that

$$
H(\Delta) = \frac{e^{-2\alpha}}{2\, k^3\, d_1} \sum_{i=1}^{d_1} \sum_{\ell, \ell'=1}^{k} \langle\!\langle \Delta, e_{i, j_{i,\ell}} - e_{i, j_{i,\ell'}} \rangle\!\rangle^2 \sum_{\ell''=1}^{k} \chi_{i, \ell, \ell', \ell''} \,,
$$

with $\chi_{i, \ell, \ell', \ell''} = \mathbb{I}\big( \sigma_i(j_{i,\ell''}) \leq \min\{\sigma_i(j_{i,\ell}), \sigma_i(j_{i,\ell'})\} \big)$. Let $Z = \sup_{\Delta \in \mathcal{B}(D)} \mathbb{E}[H(\Delta)] - H(\Delta)$ be the worst-case random deviation of $H(\Delta)$ form its mean. We prove an upper bound

on $Z$ by showing that $Z - \mathbb{E}[Z] \leq e^{-4\alpha}D^2/(64d_1d_2)$ with high probability, and $\mathbb{E}[Z] \leq 9e^{-4\alpha}D^2/(40d_1d_2)$. This proves the desired claim in Lemma A.6.

To prove the concentration of $Z$, we utilize the random utility model (RUM) theoretic interpretation of the MNL model. The random variable $Z$ depends on the random choice of alternatives $\{j_{i,\ell}\}_{i\in[d_1],\ell\in[k]}$ and the random $k$-wise ranking outcomes $\{\sigma_i\}_{i\in[d_1]}$. The random utility theory, pioneered by [2, 3, 4], tells us that the $k$-wise ranking from the MNL model has the same distribution as first drawing independent (unobserved) utilities $u_{i,\ell}$'s of the item $j_{i,\ell}$ for user $i$ according to the standard Gumbel Cumulative Distribution Function (CDF) $F(c - \Theta_{i,j_{i,\ell}})$ with $F(c) = e^{-e^{-c}}$, and then ranking the $k$ items for user $i$ according to their respective utilities. Given this definition of the MNL model, we have $\chi_{i,\ell,\ell',\ell''} = \mathbb{I}\left(u_{i,\ell''} \geq \max\{u_{i,\ell}, u_{i,\ell'}\}\right)$. Thus $Z$ is a function of independent choices of the items and their (unobserved) utilities, i.e. $Z = f(\{(j_{i,\ell}, u_{i,\ell})\}_{i\in[d_1],\ell\in[k]})$. Let $x_{i,\ell} = (j_{i,\ell}, u_{i,\ell})$ and write $H(\Delta)$ as $H(\Delta, \{x_{i,\ell}\}_{i\in[d_1],\ell\in[k]})$. This allows us to bound the difference and apply McDiarmid's tail bound. Note that for any $i \in [d_1]$, $\ell \in [k]$, $x_{1,1}, \ldots, x_{d_1,k}$, and $x'_{i,\ell}$,

$$\left| f\left(x_{1,1}, \ldots, x_{i,\ell}, \ldots, x_{d_1,k}\right) - f\left(x_{1,1}, \ldots, x'_{i,\ell}, \ldots, x_{d_1,k}\right) \right|$$

$$= \left| \sup_{\Delta \in \mathcal{B}(D)} \left(\mathbb{E}\left[H(\Delta)\right] - H(\Delta, x_{1,1}, \ldots, x_{i,\ell}, \ldots, x_{d_1,k})\right) - \sup_{\Delta \in \mathcal{B}(D)} \left(\mathbb{E}\left[H(\Delta)\right] - H(\Delta, x_{1,1}, \ldots, x'_{i,\ell}, \ldots, x_{d_1,k})\right) \right|$$

$$\leq \sup_{\Delta \in \mathcal{B}(D)} \left| H(\Delta, x_{1,1}, \ldots, x_{i,\ell}, \ldots, x_{d_1,k}) - H(\Delta, x_{1,1}, \ldots, x'_{i,\ell}, \ldots, x_{d_1,k}) \right|$$

$$\overset{(a)}{\leq} \frac{e^{-2\alpha}}{2\,k^3\,d_1} \sup_{\Delta \in \mathcal{B}(D)} \left\{ 2 \sum_{\ell'\in[k]} \langle\!\langle \Delta, e_{i,j_{i,\ell}} - e_{i,j_{i,\ell'}} \rangle\!\rangle^2 \sum_{\ell''=1}^{k} \chi_{i,\ell,\ell',\ell''} + \sum_{\ell',\ell''\in[k]} \langle\!\langle \Delta, e_{i,j_{i,\ell'}} - e_{i,j_{i,\ell''}} \rangle\!\rangle^2 \chi_{i,\ell',\ell'',\ell} \right\}$$

$$\overset{(b)}{\leq} \frac{8\alpha^2 e^{-2\alpha}}{k^3\,d_1} \left\{ 2 \sum_{\ell'\in[k]\setminus\{\ell\}} \sum_{\ell''=1}^{k} \chi_{i,\ell,\ell',\ell''} + \sum_{\ell',\ell''\in[k],\ell'\neq\ell''} \chi_{i,\ell',\ell'',\ell} \right\}$$

$$\leq \frac{16\alpha^2 e^{-2\alpha}}{k\,d_1} ,$$

where $(a)$ follows because for a fixed $i$ and $\ell$, the random variable $x_{i,\ell} = (j_{i,\ell}, u_{i,\ell})$ can appear in three terms, i.e. $\sum_{\ell',\ell''}\langle\!\langle \Delta, e_{i,j_{i,\ell}} - e_{i,j_{i,\ell'}} \rangle\!\rangle^2 \chi_{i,\ell,\ell',\ell''} + \sum_{\ell',\ell''}\langle\!\langle \Delta, e_{i,j_{i,\ell'}} - e_{i,j_{i,\ell}} \rangle\!\rangle^2 \chi_{i,\ell',\ell,\ell''} + \sum_{\ell',\ell''}\langle\!\langle \Delta, e_{i,j_{i,\ell'}} - e_{i,j_{i,\ell''}} \rangle\!\rangle^2 \chi_{i,\ell',\ell'',\ell}$, and $(b)$ follows because $|\Delta_{ij}| \leq 2\alpha$ for all $i, j$ since $\Delta \in \mathcal{B}(D)$. The last inequality follows because in the worst case, $\sum_{\ell'\in[k]\setminus\{\ell\}}\sum_{\ell''=1}^{k}\chi_{i,\ell,\ell',\ell''} \leq k(k-1)/2$ and $\sum_{\ell',\ell''\in[k],\ell'\neq\ell''}\chi_{i,\ell',\ell'',\ell} \leq k(k-1)$. This holds with equality if $\sigma_i(j_{i,\ell}) = k$ and $\sigma_i(j_{i,\ell}) = 1$, respectively. By bounded differences inequality, we have

$$\mathbb{P}\left\{ Z - \mathbb{E}\left[Z\right] \geq t \right\} \leq \exp\left( -\frac{k^2\,d_1^2\,t^2}{2^7\,\alpha^4 e^{-4\alpha}d_1 k} \right),$$

It follows that for the choice of $t = e^{-4\alpha}D^2/(64d_1d_2)$,

$$\mathbb{P}\left\{ Z - \mathbb{E}\left[Z\right] \geq \frac{e^{-4\alpha}D^2}{64d_1d_2} \right\} \leq \exp\left( -\frac{e^{-4\alpha}kD^4}{2^{19}\alpha^4 d_1 d_2^2} \right).$$

We are left to prove the upper bound on $\mathbb{E}[Z]$ using symmetrization and contraction. Define random variables

$$Y_{i,\ell,\ell',\ell''}(\Delta) \equiv (\Delta_{i,j_{i,\ell}} - \Delta_{i,j_{i,\ell'}})^2 \chi_{i,\ell,\ell',\ell''} , \tag{46}$$

where the randomness is in the choice of alternatives $j_{i,\ell}, j_{i,\ell'}$, and $j_{i,\ell''}$, and the outcome of the comparisons of those three alternatives.

The main challenge in applying the symmetrization to $\sum_{\ell,\ell',\ell''\in[k]} Y_{i,\ell,\ell',\ell''}(\Delta)$ is that we need to partition the summation over the set $[k] \times [k] \times [k]$ into subsets of independent random variables, such that we can apply the standard symmetrization argument. to this end, we prove in the following lemma a a generalization of the well-known problem of scheduling a

round robin tournament to a tournament of matches involving three teams each. No teams are present in more than one triple in a single round, and we want to minimize the number of rounds to cover all combination of triples are matched. For example, when there are $k = 6$ teams, there is a simple construction of such a tournament: $T_1 = \{(1,2,3),(4,5,6)\}$, $T_2 = \{1,2,4),(3,5,6)\}$, $T_3 = \{(1,2,5),(3,4,6)\}$, $T_4 = \{(1,2,6),(3,4,5)\}$, $T_5 = \{(1,3,4),(2,5,6)\}$, $T_6 = \{(1,3,5),(2,4,6)\}$, $T_7 = \{(1,3,6),(2,4,5)\}$, $T_8 = \{(1,4,5),(2,3,6)\}$, $T_9 = \{(1,4,6),(2,3,5)\}$, $T_{10} = \{(1,5,6),(2,3,4)\}$. This is a perfect scheduling of a tournament with three teams in each match. For a general $k$, the following lemma provides a construction with $O(k^2)$ rounds.

**Lemma A.7.** *There exists a partition $(T_1, \ldots, T_N)$ of $[k] \times [k] \times [k]$ for some $N \leq 24k^2$ such that $T_a$'s are disjoint subsets of $[k] \times [k] \times [k]$, $\bigcup_{a \in [N]} T_a = [k] \times [k] \times [k]$, $|T_a| \leq \lfloor k/3 \rfloor$ and for any $a \in [N]$ the set of random variables in $T_a$ satisfy*

$$\{Y_{i,\ell,\ell',\ell''}\}_{i \in [d_1],(\ell,\ell',\ell'') \in T_a} \text{ are mutually independent } .$$

Now, we are ready to partition the summation.

$$
\begin{aligned}
\mathbb{E}[Z] &= \frac{e^{-2\alpha}}{2\,k^3\,d_1} \mathbb{E}\Big[ \sup_{\Delta \in \mathcal{B}(D)} \sum_{i \in [d_1]} \sum_{\ell,\ell',\ell'' \in [k]} \big\{ \mathbb{E}[Y_{i,\ell,\ell',\ell''}(\Delta)] - Y_{i,\ell,\ell',\ell''}(\Delta) \big\} \Big] \\
&= \frac{e^{-2\alpha}}{2\,k^3\,d_1} \mathbb{E}\Big[ \sup_{\Delta \in \mathcal{B}(D)} \sum_{i \in [d_1]} \sum_{a \in [N]} \sum_{(\ell,\ell',\ell'') \in T_a} \big\{ \mathbb{E}[Y_{i,\ell,\ell',\ell''}(\Delta)] - Y_{i,\ell,\ell',\ell''}(\Delta) \big\} \Big] \\
&\leq \frac{e^{-2\alpha}}{2\,k^3\,d_1} \sum_{a \in [N]} \mathbb{E}\Big[ \sup_{\Delta \in \mathcal{B}(D)} \sum_{i \in [d_1]} \sum_{(\ell,\ell',\ell'') \in T_a} \big\{ \mathbb{E}[Y_{i,\ell,\ell',\ell''}(\Delta)] - Y_{i,\ell,\ell',\ell''}(\Delta) \big\} \Big] \\
&\leq \frac{e^{-2\alpha}}{k^3\,d_1} \sum_{a \in [N]} \mathbb{E}\Big[ \sup_{\Delta \in \mathcal{B}(D)} \sum_{i \in [d_1]} \sum_{(\ell,\ell',\ell'') \in T_a} \xi_{i,\ell,\ell',\ell''} Y_{i,\ell,\ell',\ell''}(\Delta) \Big] \\
&= \frac{e^{-2\alpha}}{k^3\,d_1} \sum_{a \in [N]} \mathbb{E}\Big[ \sup_{\Delta \in \mathcal{B}(D)} \sum_{i \in [d_1]} \sum_{(\ell,\ell',\ell'') \in T_a} \xi_{i,\ell,\ell',\ell''} (\Delta_{i,j_{i,\ell}} - \Delta_{i,j_{i,\ell'}})^2 \chi_{i,\ell,\ell',\ell''} \Big] \quad (47)
\end{aligned}
$$

where the first inequality follows from the fact that sum of the supremum if no less than the supremum of the sum, and the second inequality follows from standard symmetrization argument applied to independent random variables $\{Y_{i,\ell,\ell',\ell''}(\Delta)\}_{i \in [d_1],(\ell,\ell',\ell'') \in T_a}$ with i.i.d. Rademacher random variables $\xi_{i,\ell,\ell',\ell''}$'s. Since $(\Delta_{i,j_{i,\ell}} - \Delta_{i,j_{i,\ell'}})^2 \chi_{i,\ell,\ell',\ell''} \leq 4\alpha|\Delta_{i,j_{i,\ell}} - \Delta_{i,j_{i,\ell'}}|\chi_{i,\ell,\ell',\ell''}$, we have by the Ledoux-Talagrand contraction inequality that

$$
\begin{aligned}
&\mathbb{E}\Big[ \sup_{\Delta \in \mathcal{B}(D)} \sum_{i \in [d_1]} \sum_{(\ell,\ell',\ell'') \in T_a} \xi_{i,\ell,\ell',\ell''} (\Delta_{i,j_{i,\ell}} - \Delta_{i,j_{i,\ell'}})^2 \chi_{i,\ell,\ell',\ell''} \Big] \\
&\leq 8\alpha \mathbb{E}\Big[ \sup_{\Delta \in \mathcal{B}(D)} \sum_{i \in [d_1]} \sum_{(\ell,\ell',\ell'') \in T_a} \xi_{i,\ell,\ell',\ell''} \chi_{i,\ell,\ell',\ell''} \langle\!\langle \Delta, e_i(e_{j_{i,\ell}} - e_{j_{i,\ell'}})^T \rangle\!\rangle \Big] \quad (48)
\end{aligned}
$$

Applying Hölder's inequality, we get that

$$
\begin{aligned}
&\Big| \sum_{i \in [d_1]} \sum_{(\ell,\ell',\ell'') \in T_a} \xi_{i,\ell,\ell',\ell''} \chi_{i,\ell,\ell',\ell''} \langle\!\langle \Delta, e_i(e_{j_{i,\ell}} - e_{j_{i,\ell'}})^T \rangle\!\rangle \Big| \\
&\leq \|\Delta\|_{\mathrm{nuc}} \Big\|\!\!\Big\| \sum_{i \in [d_1]} \sum_{(\ell,\ell',\ell'') \in T_a} \xi_{i,\ell,\ell',\ell''} \chi_{i,\ell,\ell',\ell''} \big( e_i(e_{j_{i,\ell}} - e_{j_{i,\ell'}})^T \big) \Big\|\!\!\Big\|_2 . \quad (49)
\end{aligned}
$$

We are left to prove that the expected value of the right-hand side of the above inequality is bounded by $C\|\Delta\|_{\mathrm{nuc}} \sqrt{kd_1 \log d / \min\{d_1, d_2\}}$ for some numerical constant $C$. For $i \in [d_1]$ and $(\ell,\ell',\ell'') \in T_a$, let $W_{i,\ell,\ell',\ell''} = \xi_{i,\ell,\ell',\ell''} \chi_{i,\ell,\ell',\ell''} \big( e_i(e_{j_{i,\ell}} - e_{j_{i,\ell'}})^T \big)$ be independent zero-mean random matrices, such that

$$\|W_{i,\ell,\ell',\ell''}\|_2 = \Big\|\!\!\Big\| \xi_{i,\ell,\ell',\ell''} \chi_{i,\ell,\ell',\ell''} \big( e_i(e_{j_{i,\ell}} - e_{j_{i,\ell'}})^T \big) \Big\|\!\!\Big\|_2 \leq \sqrt{2} ,$$

almost surely, and

$$
\begin{aligned}
\mathbb{E}[W_{i,\ell,\ell',\ell''}W_{i,\ell,\ell',\ell''}^T] &= \mathbb{E}[(e_i(e_{j_{i,\ell}} - e_{j_{i,\ell'}})^T(e_{j_{i,\ell}} - e_{j_{i,\ell'}})e_i^T)\chi_{i,\ell,\ell',\ell''}] \\
&= 2\mathbb{E}\left[\chi_{i,\ell,\ell',\ell''}\right]e_i e_i^T \\
&\preceq 2e_i e_i^T \ ,
\end{aligned}
$$

and

$$
\begin{aligned}
\mathbb{E}[W_{i,\ell,\ell',\ell''}^T W_{i,\ell,\ell',\ell''}] &= \mathbb{E}[((e_{j_{i,\ell}} - e_{j_{i,\ell'}})e_i^T e_i(e_{j_{i,\ell}} - e_{j_{i,\ell'}})^T)\chi_{i,\ell,\ell',\ell''}] \\
&\preceq \mathbb{E}[(e_{j_{i,\ell}} - e_{j_{i,\ell'}})e_i^T e_i(e_{j_{i,\ell}} - e_{j_{i,\ell'}})^T] \\
&= \frac{2}{d_2}\mathbf{I}_{d_2 \times d_2} - \frac{2}{d_2^2}\mathbb{1}\mathbb{1}^T \ .
\end{aligned}
$$

This gives

$$
\begin{aligned}
\sigma^2 &= \max\left\{\left\|\left\|\sum_{i\in[d_1]}\sum_{(\ell,\ell',\ell'')\in T_a}\mathbb{E}[W_{i,\ell,\ell',\ell''}W_{i,\ell,\ell',\ell''}^T]\right\|\right\|_2, \left\|\left\|\sum_{i\in[d_1]}\sum_{(\ell,\ell',\ell'')\in T_a}\mathbb{E}[W_{i,\ell,\ell',\ell''}^T W_{i,\ell,\ell',\ell''}]\right\|\right\|_2\right\} \\
&\leq \max\left\{2|T_a|, \frac{2d_1|T_a|}{d_2}\right\} = \frac{2d_1|T_a|}{\min\{d_1,d_2\}} \leq \frac{2d_1 k}{3\min\{d_1,d_2\}} \ ,
\end{aligned}
$$

since we have designed $T_a$'s such that $|T_a| \leq k/3$. Applying matrix Bernstein inequality [25] yields the tail bound

$$
\mathbb{P}\left\{\left\|\left\|\sum_{i\in[d_1]}\sum_{(\ell,\ell',\ell'')\in T_a}W_{i,\ell,\ell',\ell''}\right\|\right\|_2 \geq t\right\} \leq (d_1 + d_2)\exp\left(\frac{-t^2/2}{\sigma^2 + \sqrt{2}t/3}\right) \ .
$$

Choosing $t = \max\left\{\sqrt{32kd_1\log d/(3\min\{d_1,d_2\})}, (16\sqrt{2}/3)\log d\right\}$, we obtain with probability at least $1 - 2d^{-3}$,

$$
\left\|\left\|\sum_{i\in[d_1]}\sum_{(\ell,\ell',\ell'')\in T_a}W_{i,\ell,\ell',\ell''}\right\|\right\|_2 \leq \max\left\{\sqrt{\frac{32kd_1\log d}{3\min\{d_1,d_2\}}}, \frac{16\sqrt{2}\log d}{3}\right\} \ .
$$

It follows from the fact $\left\|\left\|\sum_{i\in[d_1]}\sum_{(\ell,\ell',\ell'')\in T_a}W_{i,\ell,\ell',\ell''}\right\|\right\|_2 \leq \sum_{i,(\ell,\ell',\ell'')}\||W_{i,\ell,\ell',\ell''}\||_2 \leq \sqrt{2}d_1 k/3$ that

$$
\begin{aligned}
\mathbb{E}\left[\left\|\left\|\sum_{i\in[d_1]}\sum_{(\ell,\ell',\ell'')\in T_a}W_{i,\ell,\ell',\ell''}\right\|\right\|_2\right] &\leq \max\left\{\sqrt{\frac{32kd_1\log d}{3\min\{d_1,d_2\}}}, \frac{16\sqrt{2}\log d}{3}\right\} + \frac{2\sqrt{2}d_1 k}{3d^3} \\
&\leq 2\sqrt{\frac{32kd_1\log d}{3\min\{d_1,d_2\}}} \ ,
\end{aligned}
$$

where the last inequality follows from the assumption that $(16\min\{d_1,d_2\}\log d)/(3d_1) \leq k \leq d_1^2\log d$. Substituting this in the RHS of Eq. (49), and then together with Eqs. (48) and (47), this gives the following desired bound:

$$
\begin{aligned}
\mathbb{E}[Z] &\leq \sum_{a\in[N]}\sup_{\Delta\in\mathcal{B}(D)}\frac{16\alpha e^{-2\alpha}}{k^3 d_1}\sqrt{\frac{32kd_1\log d}{3\min\{d_1,d_2\}}}\||\Delta\||_{\mathrm{nuc}} \\
&\leq \sum_{a\in[N]}\frac{e^{-4\alpha}\sqrt{2}}{16\sqrt{3}k^2 d_1 d_2}\underbrace{\left(2^{10}e^{2\alpha}\alpha d_2\sqrt{\frac{d_1\log d}{k\min\{d_1,d_2\}}}\right)}_{=\mu}\||\Delta\||_{\mathrm{nuc}} \\
&\leq \frac{9e^{-4\alpha}D^2}{40d_1 d_2} \ ,
\end{aligned}
$$

where the last inequality holds because $N \leq 4k^2$ and $\mu\||\Delta\||_{\mathrm{nuc}} \leq D^2$.

## A.6 Proof of Lemma A.7

Recall that $Y_{i,\ell,\ell',\ell''}(\Delta) = (\Delta_{i,j_{i,\ell}} - \Delta_{i,j_{i,\ell'}})^2 \chi_{i,\ell,\ell',\ell''}$, as defined in (46). From the random utility model (RUM) interpretation of the MNL model presented in Section 1, it is not difficult to show that $Y_{i,\ell,\ell',\ell''}$ and $Y_{i,\tilde{\ell},\tilde{\ell}',\tilde{\ell}''}$ are mutually independent if the two triples $(\ell, \ell', \ell'')$ and $(\tilde{\ell}, \tilde{\ell}', \tilde{\ell}'')$ do not overlap, i.e., no index is present in both triples.

Now, borrowing the terminologies from round robin tournaments, we construct a schedule for a tournament with $k$ teams where each match involve three teams. Let $T_{a,b}$ denote a set of triples playing at the same round, indexed by two integers $a \in \{3, \ldots, 2k-3\}$ and $b \in \{5, \ldots, 2k-1\}$. Hence, there are total $N = (2k-5)^2$ rounds.

Each round $(a, b)$ consists of disjoint triples and is defined as

$$T_{a,b} \equiv \left\{ (\ell, \ell', \ell'') \in [k] \times [k] \times [k] \mid \ell < \ell' < \ell'', \ell + \ell' = a, \text{ and } \ell' + \ell'' = b \right\}.$$

We need to prove that $(a)$ there is no missing triple; and $(b)$ no team plays twice in a single round. First, for any ordered triple $(\ell, \ell', \ell'')$, there exists $a \in \{3, \ldots, 2k-3\}$ and $b \in \{5, \ldots, 2k-1\}$ such that $\ell + \ell' = a$ and $\ell' + \ell'' = b$. This proves that all ordered triples are covered by the above construction. Next, given a pair $(a, b)$, no two triples in $T_{a,b}$ can share the same team. Suppose there exists two distinct ordered triples $(\ell, \ell', \ell'')$ and $(\tilde{\ell}, \tilde{\ell}', \tilde{\ell}'')$ both in $T_{a,b}$, and one of the triples are shared. Then, from the two equations $\ell + \ell' = \tilde{\ell} + \tilde{\ell}' = a$ and $\ell' + \ell'' = \tilde{\ell}' + \tilde{\ell}'' = b$, it follows that all three indices must be the same, which is a contradiction. This proves the desired claim for ordered triples.

One caveat is that we wanted to cover the whole $[k] \times [k] \times [k]$, and not just the ordered triples. In the above construction, for example, a triple $(3, 2, 1)$ does not appear. This can be resolved by simply taking all $T_{a,b}$'s from the above construction, and make 6 copies of each round, and permuting all the triples in each copy according to the same permutation over $\{1, 2, 3\}$. This increases the total rounds to $N = 6(2k-5)^2 \leq 24k^2$. Note that $|T_{a,b}| \leq \lfloor k/3 \rfloor$ since no item can be in more than one triple.

## B  Proof of estimating approximate low-rank matrices in Corollary 3.2

We follow closely the proof of a similar corollary in [22]. First fix a threshold $\tau > 0$, and set $r = \max\{j | \sigma_j(\Theta^*) > \tau\}$. With this choice of $r$, we have

$$\sum_{j=r+1}^{\min\{d_1,d_2\}} \sigma_j(\Theta^*) = \tau \sum_{j=r+1}^{\min\{d_1,d_2\}} \frac{\sigma_j(\Theta^*)}{\tau} \leq \tau \sum_{j=r+1}^{\min\{d_1,d_2\}} \left(\frac{\sigma_j(\Theta^*)}{\tau}\right)^q \leq \tau^{1-q}\rho_q.$$

Also, since $r\tau^q \leq \sum_{j=1}^r \sigma_j(\Theta^*)^q \leq \rho_q$, it follows that $\sqrt{r} \leq \sqrt{\rho_a}\tau^{-q/2}$. Using these bounds, Eq. (8) is now

$$\left\|\!\left\|\widehat{\Theta} - \Theta\right\|\!\right\|_F^2 \leq \underbrace{288\sqrt{2}c_0 e^{4\alpha} d_1 d_2 \lambda_0}_{=A} \left(\sqrt{\rho_q}\tau^{-q/2}\left\|\!\left\|\widehat{\Theta} - \Theta\right\|\!\right\|_F + \tau^{1-q}\rho_q\right).$$

With the choice of $\tau = A$, it follows after some algebra that

$$\left\|\!\left\|\widehat{\Theta} - \Theta\right\|\!\right\|_F \leq 2\sqrt{\rho_q}A^{(2-q)/2}.$$

## C  Proof of the information-theoretic lower bound in Theorem 2

The proof uses information-theoretic methods which reduces the estimation problem to a multiway hypothesis testing problem. to prove a lower bound on the expected error, it suffices to prove

$$\sup_{\Theta^* \in \Omega_\alpha} \mathbb{P}\left\{\left\|\!\left\|\widehat{\Theta} - \Theta^*\right\|\!\right\|_F^2 \geq \frac{\delta^2}{4}\right\} \geq \frac{1}{2}. \tag{50}$$

To prove the above claim, we follow the standard recipe of constructing a packing in $\Omega_\alpha$. Consider a family $\{\Theta^{(1)}, \ldots, \Theta^{(M(\delta))}\}$ of $d_1 \times d_2$ dimensional matrices contained in $\Omega_\alpha$ satisfying $\left\|\left|\Theta^{(\ell_1)} - \Theta^{(\ell_2)}\right|\right\|_{\mathrm{F}} \geq \delta$ for all $\ell_1, \ell_2, \in [M(\delta)]$. We will use $M$ to refer to $M(\delta)$ for simplify the notation. Suppose we draw an index $L \in [M(\delta)]$ uniformly at random, and we are given direct observations $\sigma_i$ as per MNL model with $\Theta^* = \Theta^{(L)}$ on a randomly chosen set of $k$ items $S_i$ for each user $i \in [d_1]$. It follows from triangular inequality that

$$\sup_{\Theta^* \in \Omega_\alpha} \mathbb{P}\left\{\left\|\left|\widehat{\Theta} - \Theta^*\right|\right\|_{\mathrm{F}}^2 \geq \frac{\delta^2}{4}\right\} \;\geq\; \mathbb{P}\left\{\widehat{L} \neq L\right\}, \tag{51}$$

where $\widehat{L}$ is the resulting best estimate of the multiway hypothesis testing on $L$. The generalized Fano's inequality gives

$$\mathbb{P}\left\{\widehat{L} \neq L | S(1), \ldots, S(d_1)\right\} \;\geq\; 1 - \frac{I(\widehat{L}; L) + \log 2}{\log M} \tag{52}$$

$$\geq\; 1 - \frac{\binom{M}{2}^{-1} \sum_{\ell_1, \ell_2 \in [M]} D_{\mathrm{KL}}(\Theta^{(\ell_1)} \| \Theta^{(\ell_2)}) + \log 2}{\log M} \,, \tag{53}$$

where $D_{\mathrm{KL}}(\Theta^{(\ell_1)} \| \Theta^{(\ell_2)})$ denotes the Kullback-Leibler divergence between the distributions of the partial rankings $\mathbb{P}\left\{\sigma_1, \ldots, \sigma_{d_1} | \Theta^{(\ell_1)}, S(1), \ldots, S(d_1)\right\}$ and $\mathbb{P}\left\{\sigma_1, \ldots, \sigma_{d_1} | \Theta^{(\ell_2)}, S(1), \ldots, S(d_1)\right\}$. The second inequality follows from a standard technique, which we repeat here for completeness. Let $\Sigma = \{\sigma_1, \ldots, \sigma_{d_1}\}$ denote the observed outcome of comparisons. Since $L$–$\Theta^{(L)}$–$\Sigma$–$\widehat{L}$ form a Markov chain, the data processing inequality gives $I(\widehat{L}; L) \leq I(\Sigma; L)$. For simplicity, we drop the conditioning on the set of alternatives $\{S(1), \ldots, S(d_1)\}$, and and let $p(\cdot)$ denotes joint, marginal, and conditional distribution of respective random variables. It follows that

$$
\begin{aligned}
I(\Sigma; L) &= \sum_{\ell \in [M], \Sigma} p(\Sigma | \ell) \frac{1}{M} \log \frac{p(\ell, \Sigma)}{p(\ell) p(\Sigma)} \\
&= \frac{1}{M} \sum_{\ell \in [M]} \sum_\Sigma p(\Sigma | \ell) \log \frac{p(\Sigma | \ell)}{\frac{1}{M} \sum_{\ell'} p(\Sigma | \ell')} \\
&\leq \frac{1}{M^2} \sum_{\ell, \ell' \in [M]} \sum_\Sigma p(\Sigma | \ell) \log \frac{p(\Sigma | \ell)}{p(\Sigma | \ell')} \\
&= \frac{1}{M^2} \sum_{\ell, \ell' \in [M]} D_{\mathrm{KL}}(\Theta^{(\ell_1)} \| \Theta^{(\ell_2)}) \,, 
\end{aligned}
\tag{54}
$$

where the first inequality follows from Jensen's inequality. To compute the KL-divergence, recall that from the RUM interpretation of the MNL model (see Section 1), one can generate sample rankings $\Sigma$ by drawing random variables with exponential distributions with mean $e^{\Theta^*_{ij}}$'s. Precisely, let $X^{(\ell)} = [X_{ij}^{(\ell)}]_{i \in [d_1], j \in S_i}$ denote the set of random variables, where $X_{ij}^{(\ell)}$ is drawn from the exponential distribution with mean $e^{-\Theta_{ij}^{(\ell)}}$. The MNL ranking follows by ordering the alternatives in each $S_i$ according to this $\{X_{ij}^{(\ell)}\}_{j \in S_i}$ by ranking the smaller ones on the top. This forms a Markov chain $L$–$X^{(L)}$–$\Sigma$, and the standard data processing inequality gives

$$D_{\mathrm{KL}}(\Theta^{(\ell_1)} \| \Theta^{(\ell_2)}) \;\leq\; D_{\mathrm{KL}}(X^{(\ell_1)} \| X^{(\ell_2)}) \tag{55}$$

$$= \sum_{i \in [d_1]} \sum_{j \in S_i} \left\{ e^{\Theta_{ij}^{(\ell_1)} - \Theta_{ij}^{(\ell_2)}} - (\Theta_{ij}^{(\ell_1)} - \Theta_{ij}^{(\ell_2)}) - 1 \right\} \tag{56}$$

$$\leq \frac{e^{2\alpha}}{4\alpha^2} \sum_{i \in [d_1]} \sum_{j \in S_i} (\Theta_{ij}^{(\ell_1)} - \Theta_{ij}^{(\ell_2)})^2 \,, \tag{57}$$

where the last inequality follows from the fact that $e^x - x - 1 \leq (e^{2\alpha}/(4\alpha^2))x^2$ for any $x \in [-2\alpha, 2\alpha]$. Taking expectation over the randomly chosen set of alternatives,

$$\mathbb{E}_{S(1), \ldots, S(d_1)}[D_{\mathrm{KL}}(\Theta^{(\ell_1)} \| \Theta^{(\ell_2)})] \;\leq\; \frac{e^{2\alpha} k}{4\alpha^2 d_2} \left\|\left|\Theta^{(\ell_1)} - \Theta^{(\ell_2)}\right|\right\|_{\mathrm{F}}^2 \,. \tag{58}$$

Combined with (53), we get that

$$\mathbb{P}\left\{\widehat{L} \neq L\right\} = \mathbb{E}_{S(1),\ldots,S(d_1)}[\mathbb{P}\left\{\widehat{L} \neq L | S(1),\ldots,S(d_1)\right\}] \tag{59}$$

$$\geq 1 - \frac{\binom{M}{2}^{-1}\sum_{\ell_1,\ell_2\in[M]}(e^{2\alpha}k/(4\alpha^2 d_2))\left\|\left\|\Theta^{(\ell_1)} - \Theta^{(\ell_2)}\right\|\right\|_{\mathrm{F}}^2 + \log 2}{\log M} , \tag{60}$$

The remainder of the proof relies on the following probabilistic packing.

**Lemma C.1.** *Let $d_2 \geq d_1 \geq 607$ be positive integers. Then for each $r \in \{1,\ldots,d_1\}$, and for any positive $\delta > 0$ there exists a family of $d_1 \times d_2$ dimensional matrices $\{\Theta^{(1)},\ldots,\Theta^{(M(\delta))}\}$ with cardinality $M(\delta) = \lfloor (1/4)\exp(rd_2/576)\rfloor$ such that each matrix is rank $r$ and the following bounds hold:*

$$\left\|\left\|\Theta^{(\ell)}\right\|\right\|_{\mathrm{F}} \leq \delta , \text{ for all } \ell \in [M] \tag{61}$$

$$\left\|\left\|\Theta^{(\ell_1)} - \Theta^{(\ell_2)}\right\|\right\|_{\mathrm{F}} \geq \delta , \text{ for all } \ell_1, \ell_2 \in [M] \tag{62}$$

$$\Theta^{(\ell)} \in \Omega_{\tilde{\alpha}} , \text{ for all } \ell \in [M] , \tag{63}$$

*with $\tilde{\alpha} = (8\delta/d_2)\sqrt{2\log d}$ for $d = (d_1 + d_2)/2$.*

Suppose $\delta \leq \alpha d_2/(8\sqrt{2\log d})$ such that the matrices in the packing set are entry-wise bounded by $\alpha$, then the above lemma implies that $\left\|\left\|\Theta^{(\ell_1)} - \Theta^{(\ell_2)}\right\|\right\|_{\mathrm{F}}^2 \leq 4\delta^2$, which gives

$$\mathbb{P}\left\{\widehat{L} \neq L\right\} \geq 1 - \frac{\frac{e^{2\alpha}k\delta^2}{\alpha^2 d_2} + \log 2}{\frac{rd}{576} - 2\log 2} \geq \frac{1}{2} ,$$

where the last inequality holds for $\delta^2 \leq (\alpha^2 d_2/(e^{2\alpha}k))((rd/1152) - 2\log 2)$. If we assume $rd \geq 3195$ for simplicity, this bound on $\delta$ can be simplified to $\delta \leq \alpha e^{-\alpha}\sqrt{r\,d_2\,d/(2304\,k)}$. Together with (50) and (51), this proves that for all $\delta \leq \min\{\alpha d_2/(8\sqrt{2\log d}), \alpha e^{-\alpha}\sqrt{r\,d_2\,d/(2304\,k)}\}$,

$$\inf_{\widehat{\Theta}} \sup_{\Theta^*\in\Omega_\alpha} \mathbb{E}\left[\left\|\left\|\widehat{\Theta} - \Theta^*\right\|\right\|_{\mathrm{F}}\right] \geq \frac{\delta}{4} .$$

Choosing $\delta$ appropriately to maximize the right-hand side finishes the proof of the desired claim.

### C.1 Proof of Lemma C.1

Following the construction in [22], we use probabilistic method to prove the existence of the desired family. We will show that the following procedure succeeds in producing the desired family with probability at least half, which proves its existence. Let $d = (d_1 + d_2)/2$, and suppose $d_2 \geq d_1$ without loss of generality. For the choice of $M' = e^{rd_2/576}$, and for each $\ell \in [M']$, generate a rank-$r$ matrix $\Theta^{(\ell)} \in \mathbb{R}^{d_1 \times d_2}$ as follows:

$$\Theta^{(\ell)} = \frac{\delta}{\sqrt{rd_2}}U(V^{(\ell)})^T\left(\mathbf{I}_{d_2 \times d_2} - \frac{1}{d_2}\mathbf{1}\mathbf{1}^T\right) , \tag{64}$$

where $U \in \mathbb{R}^{d_1 \times r}$ is a random orthogonal basis such that $U^TU = \mathbf{I}_{r \times r}$ and $V^{(\ell)} \in \mathbb{R}^{d_2 \times r}$ is a random matrix with each entry $V_{ij}^{(\ell)} \in \{-1, +1\}$ chosen independently and uniformly at random.

By construction, notice that $\left\|\left\|\Theta^{(\ell)}\right\|\right\|_{\mathrm{F}} = (\delta/\sqrt{rd_2})\left\|\left\|(V^{(\ell)})^T(\mathbf{I} - (1/d_2)\mathbf{1}\mathbf{1}^T)\right\|\right\|_{\mathrm{F}} \leq \delta$, since $\left\|\left\|V^{(\ell)}\right\|\right\|_{\mathrm{F}} = \sqrt{rd_2}$ and $(\mathbf{I} - (1/d_2)\mathbf{1}\mathbf{1}^T)$ is a projection which can only decrease the norm.

Now, consider $\left\|\left\|\Theta^{(\ell_1)} - \Theta^{(\ell_2)}\right\|\right\|_{\mathrm{F}}^2 = (\delta^2/(rd_2))\left\|\left\|(\mathbf{I} - (1/d_2)\mathbf{1}\mathbf{1}^T)(V^{(\ell_1)} - V^{(\ell_2)})\right\|\right\|_{\mathrm{F}}^2 \equiv f(V^{(\ell_1)}, V^{(\ell_2)})$ which is a function over $2rd_2$ i.i.d. random Rademacher variables $V^{(\ell_1)}$ and $V^{(\ell_2)}$ which define $\Theta^{(\ell_1)}$ and $\Theta^{(\ell_2)}$ respectively. Since $f$ is Lipschitz in the following sense, we can apply McDiarmid's concentration inequality. For all $(V^{(\ell_1)}, V^{(\ell_2)})$ and $(\widetilde{V}^{(\ell_1)}, \widetilde{V}^{(\ell_2)})$ that differ in only

one variable, say $\widetilde{V}^{(\ell_1)} = V^{(\ell_1)} + 2e_{ij}$, for some standard basis matrix $e_{ij}$, we have

$$\left| f(V^{(\ell_1)}, V^{(\ell_2)}) - f(\widetilde{V}^{(\ell_1)}, \widetilde{V}^{(\ell_2)}) \right| =$$
$$\left| \frac{\delta^2}{r\, d_2} \left\|\left\| (\mathbf{I} - \frac{1}{d_2}\mathbb{1}\mathbb{1}^T)(V^{(\ell_1)} - V^{(\ell_2)}) \right\|\right\|_F^2 - \frac{\delta^2}{r\, d_2} \left\|\left\| (\mathbf{I} - \frac{1}{d_2}\mathbb{1}\mathbb{1}^T)(V^{(\ell_1)} - V^{(\ell_2)} + 2e_{ij}) \right\|\right\|_F^2 \right| \tag{65}$$

$$= \left| \frac{\delta^2}{r\, d_2} \left\|\left\| 2(\mathbf{I} - \frac{1}{d_2}\mathbb{1}\mathbb{1}^T)e_{ij} \right\|\right\|_F^2 + \frac{\delta^2}{r\, d_2} \langle\!\langle (\mathbf{I} - \frac{1}{d_2}\mathbb{1}\mathbb{1}^T)(V^{(\ell_1)} - V^{(\ell_2)}), 2e_{ij} \rangle\!\rangle \right| \tag{66}$$

$$\leq \frac{4\,\delta^2}{r\, d_2} + \frac{\delta}{r\, d_2} \left\|\left\| (\mathbf{I} - \frac{1}{d_2}\mathbb{1}\mathbb{1}^T)(V^{(\ell_1)} - V^{(\ell_2)}) \right\|\right\|_\infty \||2e_{ij}\||_1 \tag{67}$$

$$\leq \frac{12\,\delta^2}{r\, d_2}, \tag{68}$$

where we used the fact that $(\mathbf{I} - \frac{1}{d_2}\mathbb{1}\mathbb{1}^T)(V^{(\ell_1)} - V^{(\ell_2)})$ is entry-wise bounded by four. The expectation $\mathbb{E}[f(V^{(\ell_1)}, V^{(\ell_2)})]$ is

$$\frac{\delta^2}{r\, d_2}\mathbb{E}\left[ \left\|\left\| (\mathbf{I} - \frac{1}{d_2}\mathbb{1}\mathbb{1}^T)(V^{(\ell_1)} - V^{(\ell_2)}) \right\|\right\|_F^2 \right] = \frac{2\delta^2}{r\, d_2}\mathbb{E}\left[ \left\|\left\| (\mathbf{I} - \frac{1}{d_2}\mathbb{1}\mathbb{1}^T)V^{(\ell_1)} \right\|\right\|_F^2 \right] \tag{69}$$

$$= \frac{2\delta^2}{r\, d_2}\mathbb{E}\left[ \left\|\left\| V^{(\ell_1)} \right\|\right\|_F^2 \right] - \frac{2\delta^2}{r\, d_2^2}\mathbb{E}\left[ \|\mathbb{1}^T V^{(\ell_1)}\|^2 \right] \tag{70}$$

$$= \frac{2\,\delta^2\,(d_2 - 1)}{d_2}. \tag{71}$$

Applying McDiarmid's inequality with bounded difference $12\delta^2/(rd_2)$, we get that

$$\mathbb{P}\left\{ f(V^{(\ell_1)}, V^{(\ell_2)}) \leq 2\delta^2(1 - 1/d_2) - t \right\} \leq \exp\left\{ -\frac{t^2\, r\, d_2}{144\,\delta^4} \right\}, \tag{72}$$

Since there are less than $(M')^2$ pairs of $(\ell_1, \ell_2)$, setting $t = (1 - 2/d_2)\delta^2$ and applying the union bound gives

$$\mathbb{P}\left\{ \min_{\ell_1, \ell_2 \in [M']} \left\|\left\| \Theta^{(\ell_1)} - \Theta^{(\ell_2)} \right\|\right\|_F^2 \geq \delta^2 \right\} \geq 1 - \exp\left\{ -\frac{r\, d_2}{144}\left( 1 - \frac{2}{d_2} \right)^2 + 2\log M' \right\} \geq \frac{7}{8}, \tag{73}$$

where we used $M' = \exp\{rd_2/576\}$ and $d_2 \geq 607$.

We are left to prove that $\Theta^{(\ell)}$'s are in $\Omega_{(8\delta/d_2)\sqrt{2\log d_2}}$ as defined in (7). Since we removed the mean such that $\Theta^{(\ell)}\mathbb{1} = 0$ by construction, we only need to show that the maximum entry is bounded by $(8\delta/d_2)\sqrt{2\log d_2}$. We first prove an upper bound in (75) for a fixed $\ell \in [M']$, and use this to show that there exists a large enough subset of matrices satisfying this bound. From (129), consider $(UV^T)_{ij} = \langle\!\langle u_i, v_j \rangle\!\rangle$, where $u_i \in \mathbb{R}^r$ is the first $r$ entries of a random vector drawn uniformly from the $d_2$-dimensional sphere, and $v_j \in \mathbb{R}^r$ is drawn uniformly at random from $\{-1, +1\}^r$ with $\|v_j\| = \sqrt{r}$. Using Levy's theorem for concentration on the sphere [27], we have

$$\mathbb{P}\{|\langle\!\langle u_i, v_j \rangle\!\rangle| \geq t\} \leq 2\exp\left\{ -\frac{d_2\, t^2}{8\, r} \right\}. \tag{74}$$

Notice that by the definition (129), $\max_{i,j} |\Theta_{ij}^{(\ell)}| \leq (2\delta/\sqrt{rd_2})\max_{i,j} |\langle\!\langle u_i, v_j \rangle\!\rangle|$. Setting $t = \sqrt{(32r/d_2)\log d_2}$ and taking the union bound over all $d_1 d_2$ indices, we get

$$\mathbb{P}\left\{ \max_{i,j} |\Theta_{ij}^{(\ell)}| \leq \frac{2\delta\sqrt{32\log d_2}}{d_2} \right\} \geq 1 - 2d_1 d_2 \exp\left\{ -4\log d_2 \right\} \geq \frac{1}{2}, \tag{75}$$

for a fixed $\ell \in [M']$. Consider the event that there exists a subset $S \subset [M']$ of cardinality $M = (1/4)M'$ with the same bound on maximum entry, then from (75) we get

$$\mathbb{P}\left\{ \exists S \subset [M'] \text{ such that } \left\|\left\| \Theta^{(\ell)} \right\|\right\|_\infty \leq \frac{2\delta\sqrt{32\log d_2}}{d_2} \text{ for all } \ell \in S \right\} \geq \sum_{m=M}^{M'} \binom{M'}{m}\left(\frac{1}{2}\right)^m \tag{76}$$

which is larger than half for our choice of $M < M'/2$.

## D   Proof of Theorem 3

We use similar notations and techniques as the proof of Theorem 1 in Appendix A. From the definition of $\mathcal{L}(\Theta)$ in Eq. (17), we have for the true parameter $\Theta^*$, the gradient evaluated at the true parameter is

$$\nabla\mathcal{L}(\Theta^*) \;=\; -\frac{1}{n}\sum_{i=1}^{n}(e_{u_i}e_{v_i}^T - p_i)\,, \tag{77}$$

where $p_i$ denotes the conditional probability of the MNL choice for the $i$-th sample. Precisely, $p_i = \sum_{j_1 \in S_i}\sum_{j_2 \in T_i} p_{j_1,j_2|S_i,T_i} e_{j_1} e_{j_2}^T$ where $p_{j_1,j_2|S_i,T_i}$ is the probability that the pair of items $(j_1, j_2)$ is chosen at the $i$-th sample such that $p_{j_1,j_2|S_i,T_i} \equiv \mathbb{P}\{(u_i, v_i) = (j_1, j_2)|S_i, T_i\} = e^{\Theta^*_{j_1,j_2}}/(\sum_{j_1' \in S_i, j_2' \in T_i} e^{\Theta^*_{j_1',j_2'}})$, where $(u_i, v_i)$ is the pair of items selected by the $i$-th user among the set of pairs of alternatives $S_i \times T_i$. The Hessian can be computed as

$$\frac{\partial^2 \mathcal{L}(\Theta)}{\partial\Theta_{j_1,j_2}\,\partial\Theta_{j_1',j_2'}} \;=\; \frac{1}{n}\sum_{i=1}^{n}\mathbb{I}\big((j_1,j_2)\in S_i\times T_i\big)\frac{\partial p_{j_1,j_2|S_i,T_i}}{\partial\Theta_{j_1',j_2'}} \tag{78}$$

$$= \frac{1}{n}\sum_{i=1}^{n}\mathbb{I}\big((j_1,j_2),(j_1',j_2')\in S_i\times T_i\big)\left(p_{j_1,j_2|S_i,T_i}\mathbb{I}((j_1,j_2)=(j_1',j_2')) - p_{j_1,j_2|S_i,T_i}p_{j_1',j_2'|S_i,T_i}\right)\,, \tag{79}$$

We use $\nabla^2\mathcal{L}(\Theta)\in\mathbb{R}^{d_1 d_2\times d_1 d_2}$ to denote this Hessian. Let $\Delta = \Theta^* - \widehat{\Theta}$ where $\widehat{\Theta}$ is an optimal solution to the convex optimization in (15). We introduce the following key technical lemmas.

Lemma A.1 Eq. (26)

The following lemma provides a bound on the gradient using the concentration of measure for sum of independent random matrices [25].

**Lemma D.1.** *For any positive constant $c \geq 1$ and $n \geq (4(1+c)e^{2\alpha}d_1 d_2 \log d)/\max\{d_1,d_2\}$, with probability at least $1 - 2d^{-c}$,*

$$\|\nabla\mathcal{L}(\Theta^*)\|_2 \;\leq\; \sqrt{\frac{4(1+c)e^{2\alpha}\max\{d_1,d_2\}\,\log d}{d_1\,d_2\,n}}\,. \tag{80}$$

Since we are typically interested in the regime where the number of samples is much smaller than the dimension $d_1 \times d_2$ of the problem, the Hessian is typically not positive definite. However, when we restrict our attention to the vectorized $\Delta$ with relatively small nuclear norm, then we can prove restricted strong convexity, which gives the following bound.

**Lemma D.2** (**Restricted Strong Convexity for bundled choice modeling**). *Fix any $\Theta \in \Omega_\alpha'$ and assume $(\min\{d_1,d_2\}/\min\{k_1,k_2\})\log d \leq n \leq \min\{d^5\log d, k_1 k_2 \max\{d_1^2,d_2^2\}\log d\}$. Under the random sampling model of the alternatives $\{j_{ia}\}_{i\in[n],a\in[k_1]}$ from the first set of items $[d_1]$, $\{j_{ib}\}_{i\in[n],b\in[k_1]}$ from the second set of items $[d_2]$ and the random outcome of the comparisons described in section 1, with probability larger than $1 - 2d^{-2^{25}}$,*

$$\mathrm{Vec}(\Delta)^T\,\nabla^2\mathcal{L}(\Theta)\,\mathrm{Vec}(\Delta) \;\geq\; \frac{e^{-2\alpha}}{8\,d_1\,d_2}\|\Delta\|_{\mathrm{F}}^2\,, \tag{81}$$

*for all $\Delta$ in $\mathcal{A}'$ where*

$$\mathcal{A}' = \left\{\Delta\in\mathbb{R}^{d_1\times d_2}\,\big|\,\|\Delta\|_\infty\leq 2\alpha,\ \sum_{j_1\in[d_1],j_2\in[d_2]}\Delta_{j_1 j_2}=0 \ and\ \|\Delta\|_{\mathrm{F}}^2\geq\mu'\|\Delta\|_{\mathrm{nuc}}\right\}. \tag{82}$$

*with*

$$\mu' \;\equiv\; 2^{10}\,\alpha\,d_1 d_2\sqrt{\frac{\log d}{n\,\min\{d_1,d_2\}\,\min\{k_1,k_2\}}}\,. \tag{83}$$

Building on these lemmas, the proof of Theorem 3 is divided into the following two cases. In both cases, we will show that

$$\|\Delta\|_{\mathrm{F}}^2 \;\leq\; 12\, e^{2\alpha} c_1 \lambda_1\, d_1 d_2\, \|\Delta\|_{\mathrm{nuc}}\;, \tag{84}$$

with high probability. Applying Lemma A.1 proves the desired theorem. We are left to show Eq. (84) holds.

**Case 1: Suppose** $\|\Delta\|_{\mathrm{F}}^2 \geq \mu' \|\Delta\|_{\mathrm{nuc}}$. With $\Delta = \Theta^* - \widehat{\Theta}$, the Taylor expansion yields

$$\mathcal{L}(\widehat{\Theta}) = \mathcal{L}(\Theta^*) - \langle\!\langle \nabla\mathcal{L}(\Theta^*), \Delta \rangle\!\rangle + \frac{1}{2}\mathrm{Vec}(\Delta)\nabla^2\mathcal{L}(\Theta)\mathrm{Vec}^T(\Delta), \tag{85}$$

where $\Theta = a\widehat{\Theta} + (1-a)\Theta^*$ for some $a \in [0,1]$. It follows from Lemma D.2 that with probability at least $1 - 2d^{-2^{25}}$,

$$\mathcal{L}(\widehat{\Theta}) - \mathcal{L}(\Theta^*) \;\geq\; -\|\nabla\mathcal{L}(\Theta^*)\|_2\|\Delta\|_{\mathrm{nuc}} + \frac{e^{-2\alpha}}{8\,d_1\,d_2}\|\Delta\|_{\mathrm{F}}^2\;.$$

From the definition of $\widehat{\Theta}$ as an optimal solution of the minimization, we have

$$\mathcal{L}(\widehat{\Theta}) - \mathcal{L}(\Theta^*) \;\leq\; \lambda\left(\|\Theta^*\|_{\mathrm{nuc}} - \left\|\widehat{\Theta}\right\|_{\mathrm{nuc}}\right) \;\leq\; \lambda\|\Delta\|_{\mathrm{nuc}}\;.$$

By the assumption, we choose $\lambda \geq 8\lambda_1$. In view of Lemma D.1, this implies that $\lambda \geq 2\|\nabla\mathcal{L}(\Theta^*)\|_2$ with probability at least $1 - 2d^{-3}$. It follows that with probability at least $1 - 2d^{-3} - 2d^{-2^{25}}$,

$$\frac{e^{-2\alpha}}{8d_1d_2}\|\Delta\|_{\mathrm{F}}^2 \;\leq\; \left(\lambda + \|\nabla\mathcal{L}(\Theta^*)\|_2\right)\|\Delta\|_{\mathrm{nuc}} \;\leq\; \frac{3\lambda}{2}\|\Delta\|_{\mathrm{nuc}}\;.$$

By our assumption on $\lambda \leq c_1\lambda_1$, this proves the desired bound in Eq. (84)

**Case 2: Suppose** $\|\Delta\|_{\mathrm{F}}^2 \;\leq\; \mu' \|\Delta\|_{\mathrm{nuc}}$. By the definition of $\mu$ and the fact that $c_1 \geq 128/\sqrt{\min\{k_1,k_2\}}$, it follows that $\mu' \leq 12\, e^{2\alpha} c_1\lambda_1\, d_1 d_2$, and we get the same bound as in Eq. (84).

### D.1 Proof of Lemma D.1

Define $X_i = -(e_{u_i}e_{v_i}^T - p_i)$ such that $\nabla\mathcal{L}(\Theta^*) = (1/n)\sum_{i=1}^n X_i$, which is a sum of $n$ independent random matrices. Note that since $p_i$ is entry-wise bounded by $e^{2\alpha}/(k_1 k_2)$,

$$\|X_i\|_2 \;\leq\; 1 + \frac{e^{2\alpha}}{\sqrt{k_1 k_2}}\;,$$

and

$$\sum_{i=1}^n \mathbb{E}[X_i X_i^T] \;=\; \sum_{i=1}^n (\mathbb{E}[e_{u_i}e_{u_i}^T] - p_i p_i^T) \tag{86}$$

$$\preceq\; \sum_{i=1}^n \mathbb{E}[e_{u_i}e_{u_i}^T] \tag{87}$$

$$\preceq\; \frac{e^{2\alpha}\, n}{d_1}\mathbf{I}_{d_1 \times d_1}\;, \tag{88}$$

where the last inequality follows from the fact that for any given $S_i$, $u_i$ will be chosen with probability at most $e^{2\alpha}/k_1$, if it is in the set $S_i$ which happens with probability $k_1/d_1$. Therefore,

$$\left\|\sum_{i=1}^n \mathbb{E}[X_i X_i^T]\right\|_2 \;\leq\; \frac{e^{2\alpha}\, n}{d_1}\;. \tag{89}$$

Similarly,

$$\left\|\sum_{i=1}^n \mathbb{E}[X_i^T X_i]\right\|_2 \;\leq\; \frac{e^{2\alpha}\, n}{d_2}\;. \tag{90}$$

Applying matrix Bernstein inequality [25], we get

$$\mathbb{P}\left\{\left\|\nabla\mathcal{L}(\Theta^*)\right\|_2 > t\right\} \leq (d_1 + d_2)\exp\left\{\frac{-n^2 t^2/2}{(e^{2\alpha}n\max\{d_1,d_2\}/(d_1 d_2)) + ((1 + (e^{2\alpha}/\sqrt{k_1 k_2}))nt/3)}\right\}, \quad (91)$$

which gives the desired tail probability of $2d^{-c}$ for the choice of

$$\begin{aligned}
t &= \max\left\{\sqrt{\frac{4(1+c)e^{2\alpha}\max\{d_1,d_2\}\log d}{d_1 d_2 n}}, \frac{4(1+c)(1+\frac{e^{2\alpha}}{\sqrt{k_1 k_2}})\log d}{3n}\right\} \\
&= \sqrt{\frac{4(1+c)e^{2\alpha}\max\{d_1,d_2\}\log d}{d_1 d_2 n}},
\end{aligned}$$

where the last equality follows from the assumption that $n \geq (4(1+c)e^{2\alpha}d_1 d_2 \log d)/\max\{d_1,d_2\}$.

### D.2 Proof of Lemma D.2

Thee quadratic form of the Hessian defined in (79) can be lower bounded by

$$\text{Vec}(\Delta)^T\,\nabla^2\mathcal{L}(\Theta)\,\text{Vec}(\Delta) \geq \underbrace{\frac{e^{-2\alpha}}{2\,k_1^2\,k_2^2\,n}\sum_{i=1}^{n}\sum_{j_1,j_1'\in S_i}\sum_{j_2,j_2'\in T_i}\left(\Delta_{j_1,j_2} - \Delta_{j_1',j_2'}\right)^2}_{\equiv H'(\Delta)}, \quad (92)$$

which follows from Remark A.5. To lower bound $H'(\Delta)$, we first compute the mean:

$$\begin{aligned}
\mathbb{E}[H'(\Delta)] &= \frac{e^{-2\alpha}}{2\,k_1^2\,k_2^2\,n}\sum_{i=1}^{n}\mathbb{E}\Big[\sum_{j_1,j_1'\in S_i}\sum_{j_2,j_2'\in T_i}\left(\Delta_{j_1,j_2} - \Delta_{j_1',j_2'}\right)^2\Big] \quad (93) \\
&= \frac{e^{-2\alpha}}{d_1\,d_2}\|\Delta\|_{\text{F}}^2, \quad (94)
\end{aligned}$$

where we used the fact that $\mathbb{E}[\sum_{j_1\in S_i, j_2\in T_i}\Delta_{j_1,j_2}] = (k_1 k_2/(d_1 d_2))\sum_{j_1'\in[d_1],j_2'\in[d_2]}\Delta_{j_1',j_2'} = 0$ for $\Delta\in\Omega_{2\alpha}'$ in (17).

We now prove that $H'(\Delta)$ does not deviate from its mean too much. Suppose there exists a $\Delta\in\mathcal{A}'$ defined in (82) such that Eq. (81) is violated, i.e. $H'(\Delta) < (e^{-2\alpha}/(8k_1 k_2 d_1 d_2))\|\Delta\|_{\text{F}}^2$. In this case,

$$\mathbb{E}[H'(\Delta)] - H'(\Delta) \geq \frac{7\,e^{-2\alpha}}{8d_1 d_2}\|\Delta\|_{\text{F}}^2. \quad (95)$$

We will show that this happens with a small probability. We use the same peeling argument as in Appendix A with

$$\mathcal{S}_\ell' = \left\{\Delta\in\mathbb{R}^{d_1\times d_2}\,\Big|\,\|\Delta\|_{\infty}\leq 2\alpha, \beta^{\ell-1}\mu'\leq\|\Delta\|_{\text{F}}\leq\beta^{\ell}\mu', \sum_{j_1\in[d_1],j_2\in[d_2]}\Delta_{j_1,j_2} = 0, \text{ and } \|\Delta\|_{\text{nuc}}\leq\beta^{2\ell}\mu'\right\},$$

where $\beta = \sqrt{10/9}$ and for $\ell\in\{1,2,3,\ldots\}$, and $\mu'$ is defined in (83). By the peeling argument, there exists an $\ell\in\mathbb{Z}_+$ such that $\Delta\in\mathcal{S}_\ell'$ and

$$\mathbb{E}[H'(\Delta)] - H'(\Delta) \geq \frac{7\,e^{-2\alpha}}{8d_1 d_2}\beta^{2\ell-2}(\mu')^2 \geq \frac{7\,e^{-2\alpha}}{9\,d_1 d_2}\beta^{2\ell}(\mu')^2. \quad (96)$$

Applying the union bound over $\ell\in\mathbb{Z}_+$,

$$\begin{aligned}
\mathbb{P}\left\{\exists\Delta\in\mathcal{A}',\, H'(\Delta) < \frac{e^{-2\alpha}}{8\,d_1\,d_2}\|\Delta\|_{\text{F}}^2\right\} &\leq \sum_{\ell=1}^{\infty}\mathbb{P}\left\{\sup_{\Delta\in\mathcal{S}_\ell'}\left(\mathbb{E}[H'(\Delta)] - H'(\Delta)\right) > \frac{7\,e^{-2\alpha}}{9d_1 d_2}(\beta^\ell\mu')^2\right\} \\
&\leq \sum_{\ell=1}^{\infty}\mathbb{P}\left\{\sup_{\Delta\in\mathcal{B}'(\beta^\ell\mu')}\left(\mathbb{E}[H'(\Delta)] - H'(\Delta)\right) > \frac{7e^{-2\alpha}}{9d_1 d_2}(\beta^\ell\mu')^2\right\},
\end{aligned}$$

$$(97)$$

where we define the set $\mathcal{B}'(D)$ such that $\mathcal{S}'_\ell \subseteq \mathcal{B}'(\beta^\ell \mu')$:

$$\mathcal{B}'(D) = \left\{ \Delta \in \mathbb{R}^{d_1 \times d_2} \,\middle|\, \|\Delta\|_\infty \leq 2\alpha, \|\Delta\|_F \leq D, \sum_{j_1 \in [d_1], j_2 \in [d_2]} \Delta_{j_1 j_2} = 0, \mu' \|\Delta\|_{\text{nuc}} \leq D^2 \right\}.$$

(98)

The following key lemma provides the upper bound on this probability.

**Lemma D.3.** *For* $(\min\{d_1, d_2\}/\min\{k_1, k_2\}) \log d \leq n \leq d^5 \log d$,

$$\mathbb{P}\left\{ \sup_{\Delta \in \mathcal{B}'(D)} \left( \mathbb{E}[H'(\Delta)] - H'(\Delta) \right) \geq \frac{e^{-2\alpha} D^2}{2 d_1 d_2} \right\} \leq \exp\left\{ -\frac{n \min\{k_1^2, k_2^2\} k_1 k_2 D^4}{2^{10} \alpha^4 d_1^2 d_2^2} \right\}$$

(99)

Let $\eta = \exp\left( -\frac{n k_1 k_2 \min\{k_1^2, k_2^2\}(\beta - 1.002)(\mu')^4}{2^{10} \alpha^4 d_1^2 d_2^2} \right)$. Applying the tail bound to (97), we get

$$\mathbb{P}\left\{ \exists \Delta \in \mathcal{A}', \ H'(\Delta) < \frac{e^{-2\alpha}}{8 d_1 d_2} \|\Delta\|_F^2 \right\} \leq \sum_{\ell=1}^\infty \exp\left\{ -\frac{n \, k_1 k_2 \, \min\{k_1^2, k_2^2\} \, (\beta^\ell \mu')^4}{2^{10} \alpha^4 d_1^2 d_2^2} \right\}$$

$$\overset{(a)}{\leq} \sum_{\ell=1}^\infty \exp\left\{ -\frac{n k_1 k_2 \min\{k_1^2, k_2^2\} \ell (\beta - 1.002)(\mu')^4}{2^{10} \alpha^4 d_1^2 d_2^2} \right\}$$

$$\leq \frac{\eta}{1 - \eta},$$

where $(a)$ holds because $\beta^x \geq x \log \beta \geq x(\beta - 1.002)$ for the choice of $\beta = \sqrt{10/9}$. By the definition of $\mu'$,

$$\eta = \exp\left\{ -\frac{2^{30} k_1 k_2 \max\{d_2^2, d_1^2\}(\log d)^2 (\beta - 1.002)}{n} \right\} \leq \exp\{-2^{25} \log d\},$$

where the last inequality follows from the assumption that $n \leq k_1 k_2 \max\{d_1^2, d_2^2\} \log d$, and $\beta - 1.002 \geq 2^{-5}$. Since for $d \geq 2$, $\exp\{-2^{25} \log d\} \leq 1/2$ and thus $\eta \leq 1/2$, the lemma follows by assembling the last two displayed inequalities.

### D.3  Proof of Lemma D.3

Let $Z \equiv \sup_{\Delta \in \mathcal{B}'(D)} \mathbb{E}[H'(\Delta)] - H'(\Delta)$ and consider the tail bound using McDiarmid's inequality. Note that $Z$ has a bounded difference of $(8\alpha^2 e^{-2\alpha} \max\{k_1, k_2\})/(k_1^2 k_2^2 n)$ when one of the $k_1 k_2 n$ independent random variables are changed, which gives

$$\mathbb{P}\{Z - \mathbb{E}[Z] \geq t\} \leq \exp\left( -\frac{k_1^4 k_2^4 n^2 t^2}{64 \alpha^4 e^{-4\alpha} \max\{k_1^2, k_2^2\} k_1 k_2 n} \right).$$

(100)

With the choice of $t = D^2/(4 e^{2\alpha} d_1 d_2)$, this gives

$$\mathbb{P}\left\{ Z - \mathbb{E}[Z] \geq \frac{e^{-2\alpha}}{4 d_1 d_2} D^2 \right\} \leq \exp\left( -\frac{k_1^3 k_2^3 n D^4}{2^{10} \alpha^4 d_1^2 d_2^2 \max\{k_1^2, k_2^2\}} \right).$$

(101)

We first construct a partition of the space similar to Lemma A.7. Let

$$\tilde{k} \equiv \min\{k_1, k_2\}.$$

(102)

**Lemma D.4.** *There exists a partition* $(\mathcal{T}_1, \ldots, \mathcal{T}_N)$ *of* $\{[k_1] \times [k_2]\} \times \{[k_1] \times [k_2]\}$ *for some* $N \leq 2k_1^2 k_2^2 / \tilde{k}$ *such that* $\mathcal{T}_\ell$'s *are disjoint subsets,* $\bigcup_{\ell \in [N]} \mathcal{T}_\ell = \{[k_1] \times [k_2]\} \times \{[k_1] \times [k_2]\}$, $|\mathcal{T}_\ell| \leq \tilde{k}$ *and for any* $\ell \in [N]$ *the set of random variables in* $\mathcal{T}_\ell$ *satisfy*

$$\{(\Delta_{j_{i,a}, j_{i,b}} - \Delta_{j_{i,a'}, j_{i,b'}})^2\}_{i \in [n], ((a,b),(a',b')) \in \mathcal{T}_\ell} \text{ are mutually independent }.$$

*where* $j_{i,a}$ *for* $i \in [n]$ *and* $a \in [k_1]$ *denote the* $a$-*th chosen item to be included in the set* $S_i$.

Now we prove an upper bound on $\mathbb{E}[Z]$ using the symmetrization technique. Recall that $j_{i,a}$ is independently and uniformly chosen from $[d_1]$ for $i \in [n]$ and $a \in [k_1]$. Similarly, $j_{i,b}$ is independently and uniformly chosen from $[d_1]$ for $i \in [n]$ and $b \in [k_2]$.

$$
\mathbb{E}[Z] = \frac{e^{-2\alpha}}{2\,k_1^2\,k_2^2\,n}\mathbb{E}\left[\sup_{\Delta \in \mathcal{B}'(D)} \sum_{i=1}^{n} \sum_{a,a' \in [k_1]} \sum_{b,b' \in [k_2]} \mathbb{E}\left[\left(\Delta_{j_{i,a},j_{i,b}} - \Delta_{j_{i,a'},j_{i,b'}}\right)^2\right] - \left(\Delta_{j_{i,a},j_{i,b}} - \Delta_{j_{i,a'},j_{i,b}}\right)^2\right] \tag{103}
$$

$$
\leq \frac{e^{-2\alpha}}{2\,k_1^2\,k_2^2\,n} \sum_{\ell \in [N]} \mathbb{E}\left[\sup_{\Delta \in \mathcal{B}'(D)} \sum_{i=1}^{n} \sum_{(j_1,j_2,j_1',j_2') \in \mathcal{T}_\ell} \mathbb{E}\left[\left(\Delta_{j_1,j_2} - \Delta_{j_1',j_2'}\right)^2\right] - \left(\Delta_{j_1,j_2} - \Delta_{j_1',j_2'}\right)^2\right] \tag{104}
$$

$$
\leq \frac{e^{-2\alpha}}{k_1^2\,k_2^2\,n} \sum_{\ell \in [N]} \mathbb{E}\left[\sup_{\Delta \in \mathcal{B}'(D)} \sum_{i=1}^{n} \sum_{(j_1,j_2,j_1',j_2') \in \mathcal{T}_\ell} \xi_{i,j_1,j_2,j_1',j_2'}\left(\Delta_{j_1,j_2} - \Delta_{j_1',j_2'}\right)^2\right], \tag{105}
$$

where the first inequality follows for the fact that the supremum of the sum is smaller than the sum of supremum, and the second inequality follows from standard symmetrization with i.i.d. Rademacher random variables $\xi_{i,j_1,j_2,j_1',j_2'}$'s. It follows from Ledoux-Talagrand contraction inequality that

$$
\mathbb{E}\left[\sup_{\Delta \in \mathcal{B}'(D)} \sum_{i=1}^{n} \sum_{(j_1,j_2,j_1',j_2') \in \mathcal{T}_\ell} \xi_{i,j_1,j_2,j_1',j_2'}\left(\Delta_{j_1,j_2} - \Delta_{j_1',j_2'}\right)^2\right] \tag{106}
$$

$$
\leq 8\alpha\,\mathbb{E}\left[\sup_{\Delta \in \mathcal{B}'(D)} \sum_{i=1}^{n} \sum_{(j_1,j_2,j_1',j_2') \in \mathcal{T}_\ell} \xi_{i,j_1,j_2,j_1',j_2'}\left(\Delta_{j_1,j_2} - \Delta_{j_1',j_2'}\right)\right] \tag{107}
$$

$$
\leq 8\alpha\,\mathbb{E}\left[\sup_{\Delta \in \mathcal{B}'(D)} \|\Delta\|_{\mathrm{nuc}} \left\|\sum_{i=1}^{n} \sum_{(j_1,j_2,j_1',j_2') \in \mathcal{T}_\ell} \xi_{i,j_1,j_2,j_1',j_2'}\left(e_{j_1,j_2} - e_{j_1',j_2'}\right)\right\|_2\right] \tag{108}
$$

$$
\leq \frac{8\alpha D^2}{\mu'}\mathbb{E}\left[\left\|\sum_{i=1}^{n} \sum_{(j_1,j_2,j_1',j_2') \in \mathcal{T}_\ell} \xi_{i,j_1,j_2,j_1',j_2'}\left(e_{j_1,j_2} - e_{j_1',j_2'}\right)\right\|_2\right], \tag{109}
$$

where the second inequality follows for the Hölder's inequality and the last inequality follows from $\mu'\|\Delta\|_{\mathrm{nuc}} \leq D^2$ for all $\Delta \in \mathcal{B}'(D)$. To bound the expected spectral norm of the random matrix, we use matrix Bernstein's inequality. Note that $\left\|\xi_{i,j_1,j_2,j_1',j_2'}c\right\|_2 \leq \sqrt{2}$ almost surely, $\mathbb{E}[(e_{j_1,j_2} - e_{j_1',j_2'})(e_{j_1,j_2} - e_{j_1',j_2'})^T] \preceq (2/d_1)\mathbf{I}_{d_1 \times d_1}$, and $\mathbb{E}[(e_{j_1,j_2} - e_{j_1',j_2'})^T(e_{j_1,j_2} - e_{j_1',j_2'})] \preceq (2/d_2)\mathbf{I}_{d_2 \times d_2}$. It follows that $\sigma^2 = 2n|\mathcal{T}_\ell|/\min\{d_1,d_2\}$, where $|\mathcal{T}_\ell| \leq \min\{k_1,k_2\}$. It follows that

$$
\mathbb{P}\left\{\left\|\sum_{i=1}^{n} \sum_{(j_1,j_2,j_1',j_2') \in \mathcal{T}_\ell} \xi_{i,j_1,j_2,j_1',j_2'}\left(e_{j_1,j_2} - e_{j_1',j_2'}\right)\right\|_2 > t\right\} \leq (d_1+d_2)\exp\left\{\frac{-t^2/2}{\frac{2n\min\{k_1,k_2\}}{\min\{d_1,d_2\}} + \frac{\sqrt{2}t}{3}}\right\},
$$

Choosing $t = \max\left\{\sqrt{64n(\min\{k_1,k_2\}/\min\{d_1,d_2\})\log d}, (16\sqrt{2}/3)\log d\right\}$, we obtain a bound on the spectral norm of $t$ with probability at least $1 - 2d^{-7}$. From the fact that $\left\|\sum_{i=1}^{n} \sum_{(j_1,j_2,j_1',j_2') \in \mathcal{T}_\ell} \xi_{i,j_1,j_2,j_1',j_2'}\left(e_{j_1,j_2} - e_{j_1',j_2'}\right)\right\|_2 \leq (n/\sqrt{2})\min\{k_1,k_2\}$, it follows that

$$
\mathbb{E}\left[\left\|\sum_{i=1}^{n} \sum_{(j_1,j_2,j_1',j_2') \in \mathcal{T}_\ell} \xi_{i,j_1,j_2,j_1',j_2'}\left(e_{j_1,j_2} - e_{j_1',j_2'}\right)\right\|_2\right] \tag{110}
$$

$$
\leq \max\left\{\sqrt{\frac{64\,n\,\min\{k_1,k_2\}\log d}{\min\{d_1,d_2\}}}, (16\sqrt{2}/3)\log d\right\} + \frac{2n\min\{k_1,k_2\}}{\sqrt{2}d^7} \tag{111}
$$

$$
\leq \sqrt{\frac{66\,n\,\min\{k_1,k_2\}\log d}{\min\{d_1,d_2\}}} \tag{112}
$$

which follows form the assumption that $n \min\{k_1, k_2\} \geq \min\{d_1, d_2\} \log d$ and $n \leq d^5 \log d$. Substituting this bound in (105), and (109), we get that

$$
\mathbb{E}[Z] \leq \frac{16 e^{-2\alpha} \alpha D^2}{\mu'} \sqrt{\frac{66 \log d}{n \min\{k_1, k_2\} \min\{d_1, d_2\}}} \tag{113}
$$

$$
\leq \frac{e^{-2\alpha} D^2}{4 \, d_1 d_2} \, . \tag{114}
$$

# E   Proof of the information-theoretic lower bound in Theorem 4

This proof follow closely the proof of Theorem 2 in Appendix C. We apply the generalized Fano's inequality in the same way to get Eq. (53)

$$
\mathbb{P}\left\{\widehat{L} \neq L\right\} \geq 1 - \frac{\binom{M}{2}^{-1} \sum_{\ell_1, \ell_2 \in [M]} D_{\mathrm{KL}}(\Theta^{(\ell_1)} \| \Theta^{(\ell_2)}) + \log 2}{\log M}, \tag{115}
$$

The main challenge in this case is that we can no longer directly apply the RUM interpretation to compete $D_{\mathrm{KL}}(\Theta^{(\ell_1)} \| \Theta^{(\ell_2)})$. This will result in over estimating the KL-divergence, because this approach does not take into account that we only take the top winner, out of those $k_1 k_2$ alternatives. Instead, we compute the divergence directly, and provide an appropriate bound. Let the set of $k_1$ rows and $k_2$ columns chosen in one of the $n$ sampling be $S \subset [d_1]$ and $T \subset [d_2]$ respectively. Then,

$$
D_{\mathrm{KL}}(\Theta^{(\ell_1)} \| \Theta^{(\ell_2)}) \overset{(a)}{=} \frac{n}{\binom{d_1}{k_1}\binom{d_2}{k_2}} \sum_{S,T} \sum_{\substack{i \in S \\ j \in T}} \frac{e^{\Theta_{ij}^{(\ell_1)}}}{\sum_{\substack{i' \in S \\ j' \in T}} e^{\Theta_{i'j'}^{(\ell_1)}}} \log \left( \frac{e^{\Theta_{ij}^{(\ell_1)}} \sum_{\substack{i' \in S \\ j' \in T}} e^{\Theta_{i'j'}^{(\ell_2)}}}{e^{\Theta_{ij}^{(\ell_2)}} \sum_{\substack{i' \in S \\ j' \in T}} e^{\Theta_{i'j'}^{(\ell_1)}}} \right) \tag{116}
$$

$$
\overset{(b)}{\leq} \frac{n}{\binom{d_1}{k_1}\binom{d_2}{k_2}} \sum_{S,T} \left( \sum_{i,j} \frac{e^{2\Theta_{ij}^{(\ell_1)}} \sum_{i',j'} e^{\Theta_{i'j'}^{(\ell_2)}} - e^{\Theta_{ij}^{(\ell_1)} + \Theta_{ij}^{(\ell_2)}} \sum_{i',j'} e^{\Theta_{i'j'}^{(\ell_1)}}}{e^{\Theta_{ij}^{(\ell_2)}} \left( \sum_{i',j'} e^{\Theta_{i'j'}^{(\ell_1)}} \right)^2} \right) \tag{117}
$$

$$
\overset{(c)}{\leq} \frac{n e^{2\alpha}}{k_1^2 k_2^2 \binom{d_1}{k_1}\binom{d_2}{k_2}} \sum_{S,T} \sum_{i,j} \left( e^{2\Theta_{ij}^{(\ell_1)} - \Theta_{ij}^{(\ell_2)}} \sum_{i',j'} e^{\Theta_{i'j'}^{(\ell_2)}} - e^{\Theta_{ij}^{(\ell_1)}} \sum_{i',j'} e^{\Theta_{i'j'}^{(\ell_1)}} \right) \tag{118}
$$

$$
= \frac{n e^{2\alpha}}{k_1^2 k_2^2 \binom{d_1}{k_1}\binom{d_2}{k_2}} \sum_{S,T} \left( \sum_{i',j'} e^{\Theta_{i'j'}^{(\ell_2)}} \sum_{i,j} \frac{\left( e^{\Theta_{ij}^{(\ell_1)}} - e^{\Theta_{ij}^{(\ell_2)}} \right)^2}{e^{\Theta_{ij}^{(\ell_2)}}} - \left( \sum_{i,j} (e^{\Theta_{ij}^{(\ell_1)}} - e^{\Theta_{ij}^{(\ell_2)}}) \right)^2 \right) \tag{119}
$$

$$
\overset{(d)}{\leq} \frac{n e^{4\alpha}}{k_1 k_2 \binom{d_1}{k_1}\binom{d_2}{k_2}} \sum_{S,T} \sum_{i,j} \left( e^{\Theta_{ij}^{(\ell_1)}} - e^{\Theta_{ij}^{(\ell_2)}} \right)^2 \tag{120}
$$

$$
\overset{(e)}{\leq} \frac{n e^{5\alpha}}{k_1 k_2 \binom{d_1}{k_1}\binom{d_2}{k_2}} \sum_{S,T} \sum_{i,j} \left( \Theta_{ij}^{(\ell_1)} - \Theta_{ij}^{(\ell_2)} \right)^2 \tag{121}
$$

$$
\overset{(f)}{=} \frac{n e^{5\alpha}}{d_1 d_2} \left\| \Theta_{ij}^{(\ell_1)} - \Theta_{ij}^{(\ell_2)} \right\|_{\mathrm{F}}^2 \tag{122}
$$

$$
\tag{123}
$$

Here $(a)$ is by definition of KL-distance and the fact that $S$, $T$ are chosen uniformly from all possible such sets and $(b)$ is due to the fact that $\log(x) \leq x - 1$ with $x = (e^{\Theta_{ij}^{(\ell_1)}} \sum_{i' \in S, j' \in T} e^{\Theta_{i'j'}^{(\ell_2)}})/(e^{\Theta_{ij}^{(\ell_2)}} \sum_{i' \in S, j' \in T} e^{\Theta_{i'j'}^{(\ell_1)}})$. The constants at $(c)$ is due to the fact that each element of $\Theta^{(\ell_1)}$ is upper bounded by $\alpha$ and lower bounded by $-\alpha$. We can get $(d)$ by removing the second term which is always negative, and using the bond of $\alpha$. $(e)$ is obtained because $e^x$

where $-\alpha \le x \le \alpha$ is Lipschitz continuous with Lipschitz constant $e^\alpha$. At last $(f)$ is obtained by simple counting of the occurrences of each $ij$. Thus we have,

$$\mathbb{P}\left\{\widehat{L} \neq L\right\} \ \ge \ 1 - \frac{\binom{M}{2}^{-1} \sum_{\ell_1, \ell_2 \in [M]} \frac{n e^{5\alpha}}{d_1 d_2} \left\|\left\|\Theta_{ij}^{(\ell_2)} - \Theta_{ij}^{(\ell_2)}\right\|\right\|_{\mathrm{F}}^2 + \log 2}{\log M}, \tag{124}$$

The remainder of the proof relies on the following probabilistic packing.

**Lemma E.1.** *Let $d_2 \ge d_1$ be sufficiently large positive integers. Then for each $r \in \{1, \ldots, d_1\}$, and for any positive $\delta > 0$ there exists a family of $d_1 \times d_2$ dimensional matrices $\{\Theta^{(1)}, \ldots, \Theta^{(M(\delta))}\}$ with cardinality $M(\delta) = \lfloor (1/4) \exp(r d_2/576) \rfloor$ such that each matrix is rank $r$ and the following bounds hold:*

$$\left\|\left\|\Theta^{(\ell)}\right\|\right\|_{\mathrm{F}} \ \le \ \delta \, , \ \text{for all } \ell \in [M] \tag{125}$$

$$\left\|\left\|\Theta^{(\ell_1)} - \Theta^{(\ell_2)}\right\|\right\|_{\mathrm{F}} \ \ge \ \frac{1}{2}\delta \, , \ \text{for all } \ell_1, \ell_2 \in [M] \tag{126}$$

$$\Theta^{(\ell)} \ \in \ \Omega'_{\tilde{\alpha}} \, , \ \text{for all } \ell \in [M] \, , \tag{127}$$

*with $\tilde{\alpha} = (8\delta/d_2)\sqrt{2\log d}$ for $d = (d_1 + d_2)/2$.*

Suppose $\delta \le \alpha d_2/(8\sqrt{2\log d})$ such that the matrices in the packing set are entry-wise bounded by $\alpha$, then the above lemma E.1 implies that $\left\|\left\|\Theta^{(\ell_1)} - \Theta^{(\ell_2)}\right\|\right\|_{\mathrm{F}}^2 \le 4\delta^2$, which gives

$$\mathbb{P}\left\{\widehat{L} \neq L\right\} \ \ge \ 1 - \frac{\frac{e^{5\alpha} n 4\delta^2}{d_1 d_2} + \log 2}{\frac{r d_2}{576} - 2\log 2} \ \ge \ \frac{1}{2} \, , \tag{128}$$

where the last inequality holds for $\delta^2 \le (r d_1 d_2^2/(1152 e^{5\alpha} n))$ and assuming $r d_2 \ge 1600$. Together with (128) and (126), this proves that for all $\delta \le \min\{\alpha d_2/(8\sqrt{2\log d}), r d_1 d_2^2/(1152 e^{5\alpha} n)\}$,

$$\inf_{\widehat{\Theta}} \sup_{\Theta^* \in \Omega_\alpha} \mathbb{E}\left[\left\|\left\|\widehat{\Theta} - \Theta^*\right\|\right\|_{\mathrm{F}}\right] \ \ge \ \delta/4 \, .$$

Choosing $\delta$ appropriately to maximize the right-hand side finishes the proof of the desired claim. Also by symmetry, we can apply the same argument to get similar bound with $d_1$ and $d_2$ interchanged.

### E.1 Proof of Lemma E.1

We show that the following procedure succeeds in producing the desired family with probability at least half, which proves its existence. Let $d = (d_1 + d_2)/2$, and suppose $d_2 \ge d_1$ without loss of generality. For the choice of $M' = e^{r d_2/576}$, and for each $\ell \in [M']$, generate a rank-$r$ matrix $\Theta^{(\ell)} \in \mathbb{R}^{d_1 \times d_2}$ as follows:

$$\Theta^{(\ell)} \ = \ \frac{\delta}{\sqrt{r d_2}} U(V^{(\ell)})^T \left(\mathbf{I}_{d_2 \times d_2} - \frac{\mathbb{1}^T U(V^{(\ell)})^T \mathbb{1}}{d_1 d_2} \mathbb{1}\mathbb{1}^T\right) \, , \tag{129}$$

where $U \in \mathbb{R}^{d_1 \times r}$ is a random orthogonal basis such that $U^T U = \mathbf{I}_{r \times r}$ and $V^{(\ell)} \in \mathbb{R}^{d_2 \times r}$ is a random matrix with each entry $V_{ij}^{(\ell)} \in \{-1, +1\}$ chosen independently and uniformly at random. By construction, notice that $\left\|\left\|\Theta^{(\ell)}\right\|\right\|_{\mathrm{F}} \le (\delta/\sqrt{r d_2})\left\|\left\|U(V^{(\ell)})^T\right\|\right\|_{\mathrm{F}} = \delta$.

Now, by triangular inequality, we have

$$\left\|\left\|\Theta^{(\ell_1)} - \Theta^{(\ell_2)}\right\|\right\|_{\mathrm{F}} \ \ge \ \frac{\delta}{\sqrt{r d_2}}\left\|\left\|U(V^{(\ell_1)} - V^{(\ell_2)})^T\right\|\right\|_{\mathrm{F}} - \frac{\delta \left|\mathbb{1}^T U(V^{(\ell_1)} - V^{(\ell_2)})^T \mathbb{1}\right|}{d_1 d_2 \sqrt{r d_2}}\left\|\left\|\mathbb{1}\mathbb{1}^T\right\|\right\|_{\mathrm{F}}$$

$$\ge \ \frac{\delta}{\sqrt{r d_2}} \underbrace{\left\|\left\|V^{(\ell_1)} - V^{(\ell_2)}\right\|\right\|_{\mathrm{F}}}_{A} - \frac{\delta}{\sqrt{r \, d_1 d_2^2}} \big( \underbrace{\left|\mathbb{1}^T U(V^{(\ell_1)})^T \mathbb{1}\right|}_{B} + \left|\mathbb{1}^T U(V^{(\ell_2)})^T \mathbb{1}\right| \big) \, .$$

We will prove that the first term is bounded by $A \geq \sqrt{rd_2}$ with probability at least 7/8 for all $M'$ matrices, and we will show that we can find $M$ matrices such that the second term is bounded by $B \leq 8\sqrt{2rd_2 \log(32r) \log(32d)}$ with probability at least 7/8. Together, this proves that with probability at least 3/4, there exists $M$ matrices such that

$$\left\| \Theta^{(\ell_1)} - \Theta^{(\ell_2)} \right\|_{\mathrm{F}} \geq \delta\left(1 - \sqrt{\frac{2^7 \log(32r) \log(32d)}{d_1 d_2}}\right) \geq \frac{1}{2}\delta \,,$$

for all $\ell_1, \ell_2 \in [M]$ and for sufficiently large $d_1$ and $d_2$.

Applying similar McDiarmid's inequality as Eq. (73) in Appendix C, it follows that $A^2 \geq rd_2$ with probability at least 7/8 for $M' = e^{rd_2/576}$ and a sufficiently large $d_2$.

To prove a bound on $B$, we will show that for a given $\ell$,

$$\mathbb{P}\left\{ |\mathbb{1}^T U (V^{(\ell)})^T \mathbb{1}| \leq 8\sqrt{2rd_2 \log(32r) \log(32d)} \right\} \geq \frac{7}{8} \,. \tag{130}$$

Then using the similar technique as in (76), it follows that we can find $M = (1/4)M'$ matrices all satisfying this bound and also the bound on the max-entry in (131). We are left to prove (130). We apply a series of concentration inequalities. Let $H_1$ be the event that $\{|\langle\!\langle V_i^{(\ell)}, \mathbb{1} \rangle\!\rangle| \leq \sqrt{2d_2 \log(32r)}$ for all $i \in [r]\}$. Then, applying the standard Hoeffding's inequality, we get that $\mathbb{P}\{H_1\} \geq 15/16$, where $V_i^{(\ell)}$ is the $i$-th column of $V^{(\ell)}$. We next change the variables and represent $\mathbb{1}^T U$ as $\sqrt{d_1} u^T \tilde{U}$, where $u$ is drawn uniformly at random from the unit sphere and $\tilde{U}$ is a $r$ dimensional subspace drawn uniformly at random. By symmetry, $\sqrt{d_1} u^T \tilde{U}$ have the same distribution as $\mathbb{1}^T U$. Let $H_2$ be the event that $\{|\langle\!\langle \tilde{U}_i, (V^{(\ell)})^T \mathbb{1} \rangle\!\rangle| \leq \sqrt{16r(d_2/d_1) \log(32r) \log(32d)}$ for all $i \in [d_1]\}$, where $\tilde{U}_i$ is the $i$-th row of $\tilde{U}$. Then, applying Levy's theorem for concentration on the sphere [27], we have $\mathbb{P}\{H_2|H_1\} \geq 15/16$. Finally, let $H_3$ be the event that $\{|\sqrt{d_1}\langle\!\langle u, \tilde{U}(V^{(\ell)})^T \rangle\!\rangle \mathbb{1}| \leq 8\sqrt{2rd_2 \log(32r) \log(32d)}\}$. Then, again applying Levy's concentration, we get $\mathbb{P}\{H_3|H_1, H_2\} \geq 15/16$. Collecting all three concentration inequalities, we get that with probability at least 13/16, $|\mathbb{1}^T U (V^{(\ell)})^T \mathbb{1}| \leq 8\sqrt{2rd_2 \log(32r) \log(32d)}$, which proves Eq. (130).

We are left to prove that $\Theta^{(\ell)}$'s are in $\Omega_{(8\delta/d_2)\sqrt{2 \log d_2}}$ as defined in (17). Similar to Eq. (75), applying Levy's concentration gives

$$\mathbb{P}\left\{ \max_{i,j} |\Theta_{ij}^{(\ell)}| \leq \frac{2\delta\sqrt{32 \log d_2}}{d_2} \right\} \geq 1 - 2\exp\left\{ -2 \log d_2 \right\} \geq \frac{1}{2} \,, \tag{131}$$

for a fixed $\ell \in [M']$. Then using the similar technique as in (76), it follows that there exists $M = (1/4)M'$ matrices all satisfying this bound and also the bound on $B$ in Eq. (130).