[Reviews · NeurIPS 2015]

Submitted by Assigned_Reviewer_1

The authors analyze techniques for statistical preference learning given ordered data, proving recovery guarantees when the underlying parameter matrix is low rank. The manuscript is well written and satisfactorily addresses important technical questions, but lacks sufficient empirical evaluation. The presented results depend on notions of restricted strong convexity, under fairly standard assumptions. Further, the authors provide both upper and lower bounds for their analysis, suggesting that the proposed sample complexity is close to optimal under the presented assumptions.

The authors combine models for collaborative ranking and bundle choice modeling under the rubric of collaborative preference learning. While it is clear that the statistical results on bundled choice modeling are closely related to the results on collaborative ranking, I think the transition between the two models is clunky, and the authors may have been better served be focusing on one or the other problem class.

I suggest that the authors consider additional experimental evaluation. For instance, it would have been useful to empirically compare the proposed approach to other approaches proposed in the literature to evaluate real work performance and/or evaluate the robustness of the proposed approaches to model mis-specification. Further, no experimental evaluation is presented for the bundled choice modeling problem (further emphasizing my suggestion of removing this section). I understand that the goal of the paper is mainly to present theoretical results, but the area of study is well served by empirical results for new techniques.

Minor comments: Line 124: If l is indexed starting from 1 what is S_{i,1} (since v_{i,0} does not exist)? The authors should clarify if the null definition is implicit.

The claim (6) is not obvious to me, can the authors provide some more detail?

Line 288: Note that deleting the row mean increases the rank of Theta by 1 in general. Please fix and double check the results of Fig 1, or show that the rank of the de-meaned parameter does not change.
Summary: The authors analyze techniques for statistical preference learning given ordered data, proving recovery guarantees when the underlying parameter matrix is low rank. The manuscript is well written and satisfactorily addresses important technical questions, but lacks sufficient empirical evaluation.

Submitted by Assigned_Reviewer_2

summary of paper: This paper proposes a new approach to inference for a low-rank matrix factorization model with ordinal data and another application.

It is not clear if MultiNomialLogit (MNL) models have already been applied to these particular problems and the contribution is just the inference approach (nuclear norm minimization), or if the models are contributions as well.

Regardless, the authors give performance guarantees for their approach, but do not demonstrate it on real or even synthetic data.

quality: good - The theory appears to be thorough.

The contributions obviously required substantial work. clarity: poor - Needs significant editing to address both minor issues and also major flow of ideas within sections.

Should have more discussion of implications of contributions. originality: mediocre - Low rank matrix factorization has already been addressed from many, many angles; there, the contribution appears incremental.

I am less familiar with the other application. significance: poor - The authors do not compare to any other methods (there are many well-established ones for matrix factorization), either empirically or theoretically.

Even a small improvement can have a large impact, but the reader is left to assume that this paper would have no practical and only minor theoretical impact on the problems described.

some detailed points - line 20 (abstract), Awkward/unclear/editing mistake?: "A popular discrete choice model of multinomial logit model captures..."

- line 18, we're obviously talking about personalized user preferences, but that could be more clear-the user isn't mentioned until after the problem of preference prediction, which seems backward; then, we're talking about comparisons without talking about what we're comparing.

This sequence of ideas is very convoluted. - abstract starts off with motivating applications, but then the contribution bears to only be theoretical (presenting an approach and proving a bound), with no justification as to why the theory is important, aside from it being a "natural approach" to for inference on "a popular model."

No hint of how the theory impacts application, nor mention of experiments on real or even synthetic data. - line 33: should be comma after applications: "applications, such as..." - 1st line of intro and abstract are identical; while not strictly prohibited, it disengages the reader - line 44, 53: elements in a list (i.e., (a) something, (b) something) should have some kind of parallel structure.

Here, (b) is the goal and (a) is the means.

Again, the sequences of ideas seem convoluted. - line 57: MultiNomialLogit model not capitalized consistently with abstract; needs citation - line 61: "The" should be capitalized at start of sentence - line 62: it might have been better to only use one example in the intro (the bundling is more novel, so I'd pick that); you can always add the other in the experiments/results - line 70: it's great to have your contribution highlighted - line 70: why is it natural to take this approach? - contribution: I think you're saying that MNL is already applied to ordinal low-rank matrix factorization, but that you're doing a new kind of inference for the model and applying that to two cases.

Is that right?

Or has the MNL only been applied to non-ordinal MF problems and you are applying the MNL to ordinal data for the first time? - line 76: RUM should also have a citation here - line 94: spacing issues - Sections 3 and 4, generally: the practical implications of the theorems and their corollaries are unclear.

Perhaps I need to study them in greater detail, but the authors could highlight in words, or empirically, the take home message.

The single figure does not immediately contribute to greater understanding. - line 289: RMSE comes out of nowhere.

Further, while RMSE is commonly used, it's not actually a good metric for rank-based item recommendation. - Discussion shouldn't just be a to-do list for the future.

What should the reader take home?
Summary: The contributions are mostly clear, but they are not well motivated or justified.

This paper needs to be thoroughly edited for clarity and the contributions should be compared to other methods on real or synthetic data.

Submitted by Assigned_Reviewer_3

The authors give a multinomial logit (MNL) model to describe user preferences, which (a) under a collaborative ranking context uses a low-rank matrix to capture the underlying preferences and (b) under the bundled choice modeling the low rank structure captures how pairs of items are matched. In order to prove bounds, the authors use the random utility model.

One contribution of this paper is that it provides a polynomial-time inference algorithm, though the Maximum Likelihood estimation for the general MNL model is intractable. Though there room for improvement, this paper could serve as the intermediate step to more general approaches which are more efficient.

The fact that the authors illuminate the weak spots of the model (e.g above equation 10) can actually help advance the research in this field.
Summary: This paper shows that the convex relaxation of nuclear norm minimization approach is minimax optimal. The authors give an upper bound on the resulting error with finite samples, and provide a matching lower bound.

Author Feedback
Author rebuttal: Reviewer_1:

Re: transition:
Our main contribution is the polynomial-time algorithm with provable guarantees for MNL. This is a fundamental problem with potentially a broad range of applications. We focus on two specific settings to showcase that this framework can be generally applied. We will make the transition more coherent, by explaining that we study a fundamental framework that has a broad impact on many applications of interest.

Re: experiments:
We fully acknowledge that experimental evaluations on real-world datasets justify the model, identify model mis-matches, and evaluate robustness. As a first step, our experiments answer some important questions: it confirms our theoretical findings and provides guidelines on how to choose the parameters in practice. Experiments on real-world datasets could potentially be done, and we believe is one of the most important next steps on this topic.

Re: minor comments:
In this setting, the de-meaned matrix rank does not increase by one. The reason is that the subspace spanned by the columns after de-meaning does not change, since the subtracted row-mean matrix still lies in the columns-space of the original matrix. We will clarify this point.

Reviewer_2:

Our main contribution is the first polynomial-time algorithm with provable guarantees for MNL. This is a first step in an important topic of learning the MNL model, which potentially has many applications (e.g. word embedding in natural language processing) but does not have much rigorous prior work to the best of our knowledge. Important (and closely related) next steps are verifying the model on real datasets and providing more efficient algorithms.

Re: novelty of the model:
The MNL model for recommendations has been proposed in [1] and [2]. We significantly improve upon those results by allowing general sampling. The MNL model has been applied to bundled purchases and we will add appropriate references.

Re: experiments:
We believe that verifying the model/algorithms on real-world datasets is a potentially important next step. Being a first algorithm with provable guarantees, our experiments answer some of the important questions: it confirms the theoretical findings and provides guidelines for choosing the parameters.

Re: clarity:
We thank the reviewer for detailed points. We provide clarifications below.

Re: originality and significance:
Although matrix factorization has been widely studied theoretically and empirically, learning MNL model poses new challenges. Our work extends the horizon of matrix completion by addressing (a) non-quadratic cost/likelihood functions; (b) samples correlated with multiple entries; and (c) non-additive noise. Some of the above extensions have partially been addressed, for example in 1-bit matrix completion. However our work is the first addressing all challenges in learning MNL (in full generality, as opposed to pairwise comparisons sampling).

The theoretical contribution is that we give (almost) matching upper and lower bound on the sample complexity. There is no theoretical comparisons, because there is no prior work with any provable guarantees in this setting. Instead, we prove a novel lower bound that holds for all algorithms. This comparison we believe is a strong result.

The practical contribution is that we provide an algorithm, which no other algorithm can improve much upon. This follows from our main results. Further, there is no existing matrix factorization algorithm that can learn MNL, hence we have no algorithms to compare against. Borrowing ideas from matrix factorization literature, we can design various algorithms, and our approach is the first step in this direction.

Re: detailed points:
We will incorporate all suggestions. We clarify some concerns below.

- line 70: The nuclear norm is the tightest convex surrogate for the rank. This leads to nuclear norm minimization to enforce low-rank solutions. This is explained later in Section 2.2, and we will add more references.

- contribution: MNL model is only defined over ordinal data. What is novel is that nuclear norm minimization for general MNL has not been studied (with exceptions of [1] and [2] for only pairwise data). This is a new kind of inference since the algorithm is new. We study two particular cases of sampling models (both under MNL).

- The take-home message is that, under both scenarios, nuclear norm minimization approach achieves (near) optimal performance, so there is no need to use other algorithms. It is true that other algorithms might have better dependence in the constant factor, and this is something to be evaluated empirically. However, there is no competing algorithm to compare against.

- The RMSE is used to be coherent with the main theorems. For parametric ranking models like the MNL model, it has been argued that RMSE is a good surrogate for (weighted) rank-based metric (e.g. [15]).